EMBO
Molecular Medicine

# BRCA1 and BRCA2 tumor suppressors protect against endogenous acetaldehyde toxicity

Eliana MC Tacconi[1,†], Xianning Lai[1,†], Cecilia Folio[1], Manuela Porru[2], Gijs Zonderland[1], Sophie Badie[1], Johanna Michl[1], Irene Sechi[1], Mélanie Rogier[3,4,5,6], Verónica Matía García[7], Ankita Sati Batra[8], Oscar M Rueda[8], Peter Bouwman[7], Jos Jonkers[7] (iD), Anderson Ryan[9], Bernardo Reina-San-Martin[3,4,5,6], Joannie Hui[10], Nelson Tang[10], Alejandra Bruna[8], Annamaria Biroccio[2] & Madalena Tarsounas[1,*] (iD)

## Abstract

Maintenance of genome integrity requires the functional interplay between Fanconi anemia (FA) and homologous recombination (HR) repair pathways. Endogenous acetaldehyde, a product of cellular metabolism, is a potent source of DNA damage, particularly toxic to cells and mice lacking the FA protein FANCD2. Here, we investigate whether HR-compromised cells are sensitive to acetaldehyde, similarly to FANCD2-deficient cells. We demonstrate that inactivation of HR factors BRCA1, BRCA2, or RAD51 hypersensitizes cells to acetaldehyde treatment, in spite of the FA pathway being functional. Aldehyde dehydrogenases (ALDHs) play key roles in endogenous acetaldehyde detoxification, and their chemical inhibition leads to cellular acetaldehyde accumulation. We find that disulfiram (Antabuse), an ALDH2 inhibitor in widespread clinical use for the treatment of alcoholism, selectively eliminates BRCA1/2-deficient cells. Consistently, *Aldh2* gene inactivation suppresses proliferation of HR-deficient mouse embryonic fibroblasts (MEFs) and human fibroblasts. Hypersensitivity of cells lacking BRCA2 to acetaldehyde stems from accumulation of toxic replication-associated DNA damage, leading to checkpoint activation, G2/M arrest, and cell death. Acetaldehyde-arrested replication forks require BRCA2 and FANCD2 for protection against MRE11-dependent degradation. Importantly, acetaldehyde specifically inhibits *in vivo* the growth of BRCA1/2-deficient tumors and *ex vivo* in patient-derived tumor xenograft cells (PDTCs), including those that are resistant to poly (ADP-ribose) polymerase (PARP) inhibitors. The work presented here therefore identifies acetaldehyde metabolism as a potential therapeutic target for the selective elimination of BRCA1/2-deficient cells and tumors.

**Keywords** BRCA1; BRCA2; disulfiram, acetaldehyde dehydrogenase; DNA damage; replication stress
**Subject Categories** Cancer; Pharmacology & Drug Discovery

See also: **AR Chaudhuri & A Nussenzweig** (October 2017)

## Introduction

*BRCA1* and *BRCA2* germ line mutations increase breast and ovarian cancer susceptibility in heterozygous carriers (Roy *et al*, 2012). More recently, *BRCA2* mutations have been linked to predisposition to prostate and pancreatic cancers (Sandhu *et al*, 2013; Mateo *et al*, 2015; Waddell *et al*, 2015). At cellular level, BRCA1 and BRCA2 proteins play critical functions in genome integrity. In response to DNA damage induced by exogenous agents (e.g., ionizing radiation), they initiate HR reactions for double-strand break (DSB) repair and in response to replication stress they act to protect and restart replication forks stalled at sites of DNA damage. In both settings, BRCA1 and BRCA2 promote loading of RAD51 recombinase onto single-stranded DNA generated at DSBs and stalled replication forks. In addition to their role in HR repair, BRCA1, BRCA2, and RAD51 have been assigned key functions in the FA pathway of interstrand cross-link (ICL) repair (Howlett *et al*, 2002; Domchek *et al*, 2013; Ameziane *et al*, 2015; Wang *et al*, 2015). ICLs represent one of the most deleterious types of DNA damage, known to obstruct both

1  Department of Oncology, Genome Stability and Tumorigenesis Group, The CR-UK/MRC Oxford Institute for Radiation Oncology, University of Oxford, Oxford, UK
2  Area of Translational Research, Regina Elena National Cancer Institute, Rome, Italy
3  Institut de Génétique et de Biologie Moléculaire et Cellulaire (IGBMC), Illkirch, France
4  Institut National de la Santé et de la Recherche Médicale (INSERM), U964, Illkirch, France
5  Centre National de Recherche Scientifique (CNRS), UMR7104, Illkirch, France
6  Université de Strasbourg, Illkirch, France
7  Division of Molecular Pathology and Cancer Genomics Netherlands, The Netherlands Cancer Institute, Amsterdam, The Netherlands
8  Cancer Research UK Cambridge Institute, Cambridge, UK
9  Department of Oncology, Lung Cancer Translational Science Research Group, The CR-UK/MRC Oxford Institute for Radiation Oncology, University of Oxford, Oxford, UK
10 Department of Chemical Pathology and Paediatrics, Faculty of Medicine, Chinese University of Hong Kong, Shatin, Hong Kong, China
   *Corresponding author. Tel: +44 1865 617319; E-mail: madalena.tarsounas@oncology.ox.ac.uk
   †These authors contributed equally to this work

replication and transcription (Kottemann & Smogorzewska, 2013). Consistent with this role, *BRCA1/2*-mutated cells and tumors are hypersensitive to ICL-inducing chemo-therapeutic agents, including cisplatin and mitomycin C (Deans & West, 2011).

The eukaryotic genome is under continuous genotoxic attack from cell-intrinsic sources, one of the most potent being endogenous aldehydes. Acetaldehyde, an intermediate in the metabolic processing of alcohol, inflicts DNA damage particularly in the form of base damage, DNA–protein crosslinks, and ICLs (Lorenti Garcia *et al*, 2009). Highly reactive aldehydes are metabolized to less toxic substrates by at least 19 different aldehyde dehydrogenase (ALDH) enzymes (Koppaka *et al*, 2012). ALDH2 in particular has been implicated in the breakdown of acetaldehyde to acetate, an obligatory step in alcohol metabolism. Disulfiram (Antabuse) is an alcohol-aversive drug metabolized to products that inhibit ALDH1A1 and ALDH2 (Chen *et al*, 2014). By inhibiting ALDH activity, disulfiram causes acetaldehyde accumulation in the blood, culminating in unpleasant and potentially serious physiological symptoms including flushing, nausea, anxiety, blurred vision, and difficulty breathing, which are intended as a deterrent to prevent alcohol consumption.

The discovery that FANCD2-deficient cells are hypersensitive to acetaldehyde provided the first indication that acetaldehyde-induced DNA damage requires the FA pathway for accurate repair (Langevin *et al*, 2011). BRCA1 and BRCA2 act in concert with the FA pathway during ICL repair to stabilize replication forks stalled at ICL sites and/or to repair resulting DNA lesions (Ceccaldi *et al*, 2016; Michl *et al*, 2016b). However, FANCD2 plays unique roles in BRCA1- or BRCA2-deficient cells by limiting replication and providing an independent mechanism for fork protection (Kais *et al*, 2016; Michl *et al*, 2016a). Thus, FANCD2 inactivation is lethal to BRCA1/2-deficient cells and tumors.

In spite of the significant functional interactions between FANCD2 and BRCA1/2 in DNA repair and replication, whether BRCA1 and BRCA2 are also required in response to acetaldehyde-induced DNA damage similarly to FANCD2 has not yet been investigated. Here, we demonstrate for the first time the striking and specific vulnerability of BRCA1/2-deficient human and mouse cells to acetaldehyde. Disulfiram is also toxic in this context, as a result of ALDH2 inhibition. We provide evidence that replication-associated DNA damage and MRE11-dependent fork degradation trigger acetaldehyde sensitivity in human cells lacking BRCA2. This further explains tumor growth inhibition by acetaldehyde in BRCA1/2-deficient tumors, including those that have acquired resistance to PARP inhibitors. Therefore, our data provide a rationale for the use of drugs that increase endogenous acetaldehyde in treatment of BRCA1/2-compromised cancers, in response to the need for novel effective therapies targeting this tumor subset.

## Results

### HR-deficient cells are hypersensitive to acetaldehyde

Mouse and chicken cells lacking FANCD2 are hypersensitive to exogenous acetaldehyde (Langevin *et al*, 2011), consistent with its ability to inflict DNA damage through DNA crosslinking (Lorenti Garcia *et al*, 2009). We thus investigated whether BRCA1 and BRCA2 deficiencies could also predispose cells to acetaldehyde

sensitivity. First, we attempted to recapitulate acetaldehyde sensitivity in human colorectal adenocarcinoma DLD1 cells in which we deleted *FANCD2* gene using the CRISPR/Cas9 system (Michl *et al*, 2016a). This deletion sensitized cells not only to cisplatin (Fig EV1A), a crosslinking agent in clinical use (Deans & West, 2011), but also to acetaldehyde treatment (Fig EV1B). Next, we investigated a potential role for BRCA2 in the cellular response to acetaldehyde by evaluating the impact of acetaldehyde on the viability of *BRCA2*-deleted DLD1 human cells (Horizon Discovery; Fig 1A). The PARP inhibitor olaparib was used as an additional control, based on its established ability to kill BRCA2-deficient cells (Bryant *et al*, 2005; Farmer *et al*, 2005). Incubation in the presence of acetaldehyde led to highly specific reduction in the viability of BRCA2-deficient DLD1 cells, comparable to the effect of olaparib treatment. Acetaldehyde also caused a significant reduction in survival of DLD1 cells lacking BRCA2 using clonogenic assays, in which olaparib and cisplatin were used as controls (Appendix Fig S1A–C). Importantly, acetaldehyde was toxic to BRCA2-deleted cells in spite of the FA pathway remaining active, as demonstrated by FANCD2 ubiquitylation (Fig EV1C).

Using RAD51 accumulation into nuclear foci as readout for HR activation, we addressed whether HR is required for the repair of DNA damage induced by acetaldehyde. We observed a marked induction of RAD51 foci in acetaldehyde-treated BRCA2-proficient DLD1 cells (Fig EV2A and B). Thus, acetaldehyde treatment invokes HR repair, which could explain the reduced survival of BRCA2-deleted cells in the presence of this compound.

To address whether acetaldehyde toxicity can be extended to other cell lines, we examined the response to this compound in human H1299 non-small cell lung carcinoma cells expressing a doxycycline (DOX)-inducible BRCA2 shRNA (BRCA2sh[DOX]; Fig 1B). DOX addition inhibited BRCA2 expression assessed by immunoblotting. We observed a profound reduction in the viability of BRCA2-deficient H1299 cells upon treatment with acetaldehyde, similar to the effect of olaparib.

Acetaldehyde was also toxic to H1299 human cells expressing a DOX-inducible shRNA against BRCA1 (BRCA1sh[DOX]; Fig 1C). Olaparib sensitivity characteristic of these cells was used as a control. Further supporting the effect on cells lacking BRCA1 tumor suppressor, acetaldehyde also targeted specifically *Brca1*[F/−] MEFs in which *Brca1*-gene was deleted using Cre recombinase (Appendix Fig S2A). In contrast, MEFs lacking 53BP1, a DNA damage response factor known to promote non-homologous end joining (NHEJ; Bouwman *et al*, 2010; Bunting *et al*, 2010), were not affected by acetaldehyde treatment (Appendix Fig S2B). Moreover, RAD51 depletion in DLD1 and H1299 human cells led to acetaldehyde hypersensitivity to a similar extent as olaparib and cisplatin (Appendix Fig S3A and B). These results clearly demonstrate specific acetaldehyde toxicity to HR-compromised cells.

### Disulfiram, an ALDH inhibitor, targets BRCA1- and BRCA2-deficient cells

Acetaldehyde is a product of physiological cell metabolism (Fig 2A), processed to acetate by ALDH enzymes, among which ALDH2 is best-characterized in human cells (Chen *et al*, 2014). Disulfiram, an ALDH inhibitor with high specificity for ALDH1A1 and ALDH2, is known to increase endogenous acetaldehyde levels, which

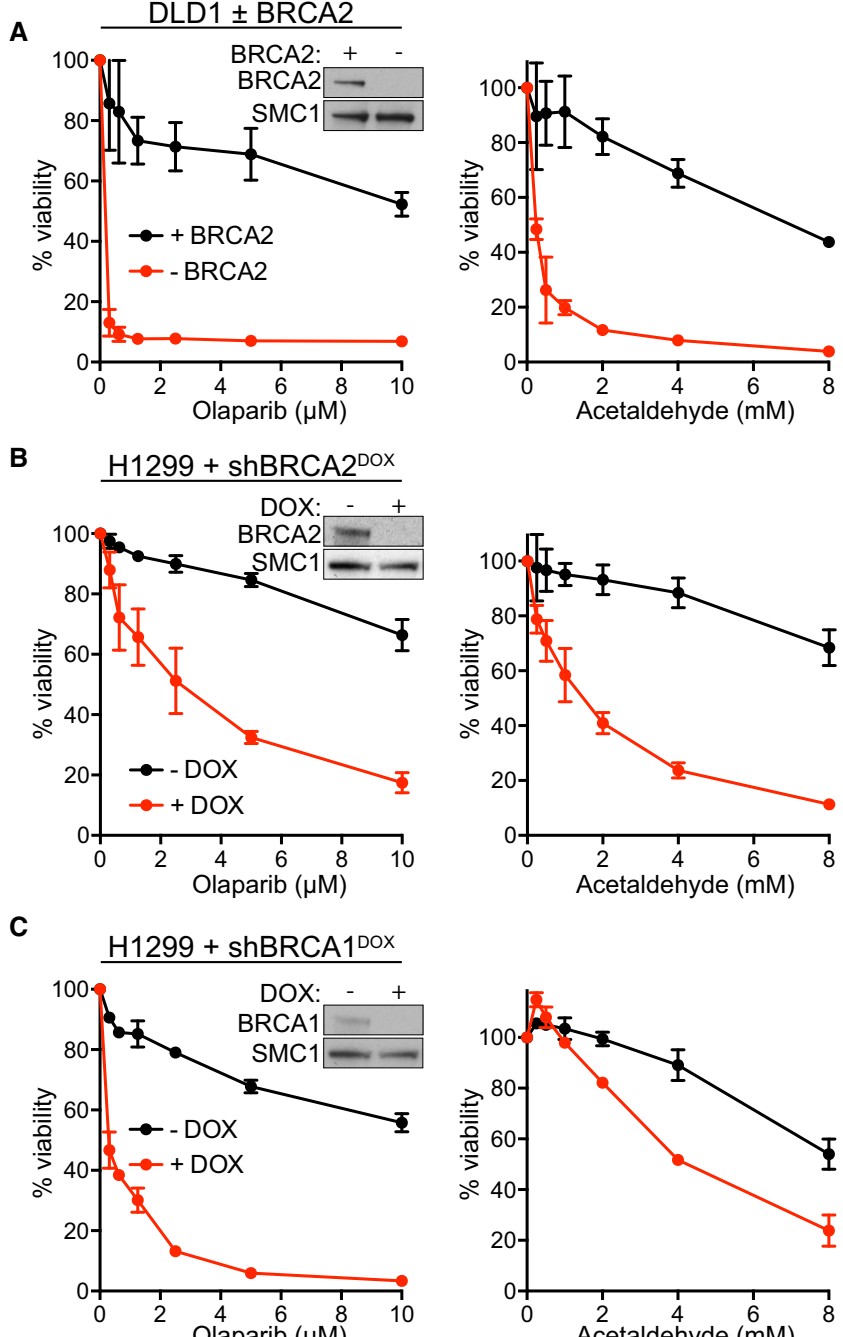

**Figure 1. BRCA1- and BRCA2-deficient human cells are hypersensitive to acetaldehyde.**

A    BRCA2-proficient (+BRCA2) or BRCA2-deficient (−BRCA2) human DLD1 cells were incubated with the indicated concentrations of olaparib or acetaldehyde for 6 days before processing for dose-dependent viability assays. Graphs are representative of three independent experiments, each performed in triplicate. Error bars represent SD of triplicate values obtained from a single experiment. Cell extracts prepared at the time of acetaldehyde addition were immunoblotted as indicated. SMC1 was used as a loading control.

B, C   Human H1299 cells expressing DOX-inducible BRCA2 or BRCA1 shRNAs were grown in the presence or absence of DOX and incubated with the indicated concentrations of olaparib or acetaldehyde for 6 days, before processing for dose-dependent viability assays. Graphs are representative of three independent experiments, each performed in triplicate. Error bars represent SD of triplicate values obtained from a single experiment. Cell extracts prepared at the time of acetaldehyde addition were immunoblotted as indicated. SMC1 was used as a loading control. DOX, doxycycline.

represents the basis of its clinical use as an alcohol deterrent (Koppaka *et al*, 2012). In a viability screen for FDA-approved drugs that kill specifically BRCA2-deficient cells, we identified disulfiram

among the high-scoring hits (our unpublished results). Moreover, we observed elevated levels of RAD51 foci upon disulfiram treatment of BRCA2-deleted DLD1 cells (Fig EV2A and B), indicating

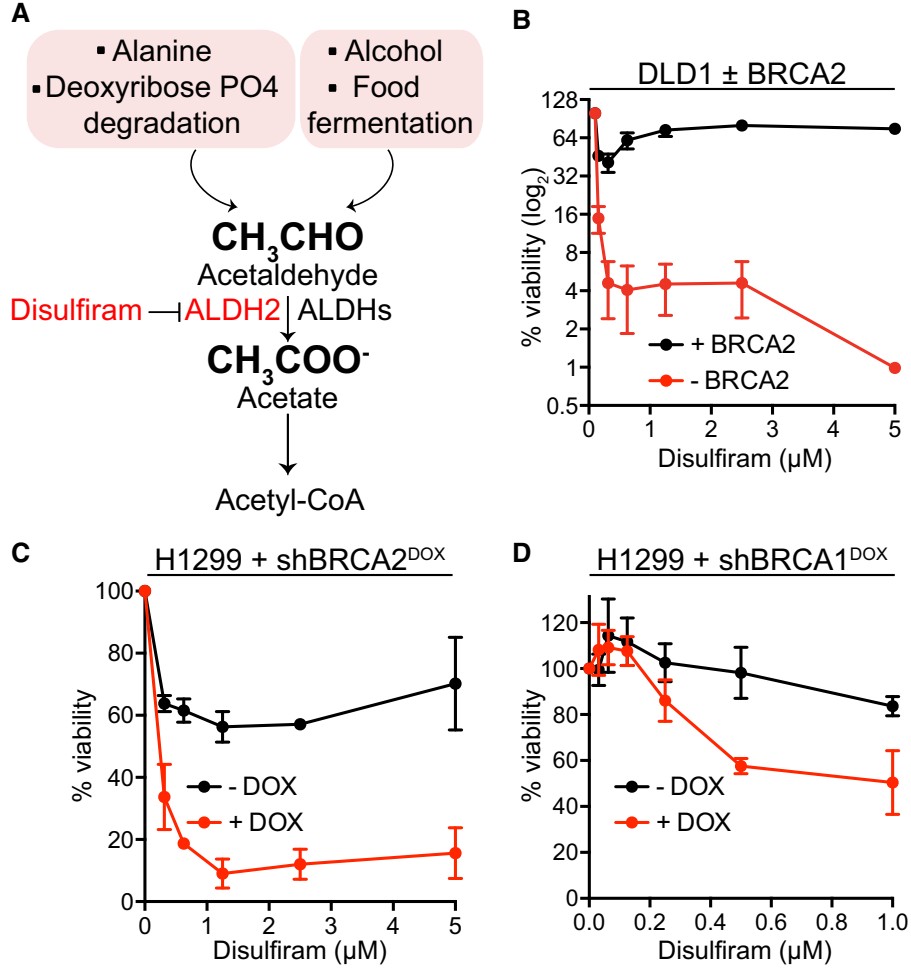

**Figure 2. BRCA1- and BRCA2-deficient human cells are hypersensitive to disulfiram.**

A    Schematic representation of sources of acetaldehyde and its cellular catabolism. ALDH2, a disulfiram target, is also indicated.
B    BRCA2-proficient (+BRCA2) or BRCA2-deficient (−BRCA2) human DLD1 cells were incubated with the indicated concentrations of disulfiram for 6 days, before processing for dose-dependent viability assays. Graphs are representative of three independent experiments, each performed in triplicate. Error bars represent SD of triplicate values obtained from a single experiment.
C, D    Human H1299 cells expressing DOX-inducible BRCA2 or BRCA1 shRNAs were grown in the presence or absence of DOX and incubated with the indicated concentrations of disulfiram for 6 days, before processing for dose-dependent viability assays. Graphs are representative of three independent experiments, each performed in triplicate. Error bars represent SD of triplicate values obtained from a single experiment. DOX, doxycycline.

that DNA damage elicited by this drug requires HR repair. These results provided a strong rationale for investigating the impact of disulfiram on the viability of human cells lacking BRCA2. Disulfiram treatment led to a striking reduction in the viability of the BRCA2-deficient DLD1 cells compared to wild-type control cells (Fig 2B and Appendix Fig S1D). FACS-based ALDEFLUOR™ assays (Garaycoechea *et al*, 2012) showed similar inhibition of ALDH activity by disulfiram in BRCA2-proficient and BRCA2-deficient DLD1 cells (Fig EV3). This excludes the possibility that the marked sensitivity of BRCA2-deficient cells to disulfiram could be due to more effective ALDH inhibition in these cells.

Incubation of BRCA2sh$^{DOX}$ H1299 cells in the presence of disulfiram also led to specific elimination of BRCA2-deficient (+DOX) relative to BRCA2-proficient (−DOX) cells (Fig 2C). Paradoxically, we observed an upturn in cell survival by increasing disulfiram concentration. This could be due to drug aggregate assembly at high

concentrations, which cannot penetrate the cellular membrane and therefore lower drug efficiency.

Furthermore, BRCA1 depletion using a DOX-inducible shRNA (BRCA1sh$^{DOX}$) in H1299 cells elicited disulfiram sensitivity (Fig 2D). The striking and specific disulfiram toxicity to cells lacking BRCA1 or BRCA2 recapitulates the effects of exogenous acetaldehyde.

## Acute replication stress induced by acetaldehyde and disulfiram in BRCA2-deficient cells

The genotoxic potential of acetaldehyde is mediated in part by its ability to cause ICLs (Lorenti Garcia *et al*, 2009), which arrest replication and inflict replication-associated DSBs. We therefore investigated the possibility that acetaldehyde accumulation, either directly administered or indirectly, mediated by disulfiram addition, could trigger replication stress. RPA sub-nuclear foci mark regions of

exposed single-stranded DNA and are commonly used as a readout for replication stress (Zeman & Cimprich, 2014). Indeed, immunofluorescence (IF) analysis of RPA foci revealed a marked induction

specifically in BRCA2-deficient DLD1 cells upon treatment with either 10 μM disulfiram or 4 mM acetaldehyde (Fig 3A and B). In contrast, treatment of BRCA2-proficient cells with either compound

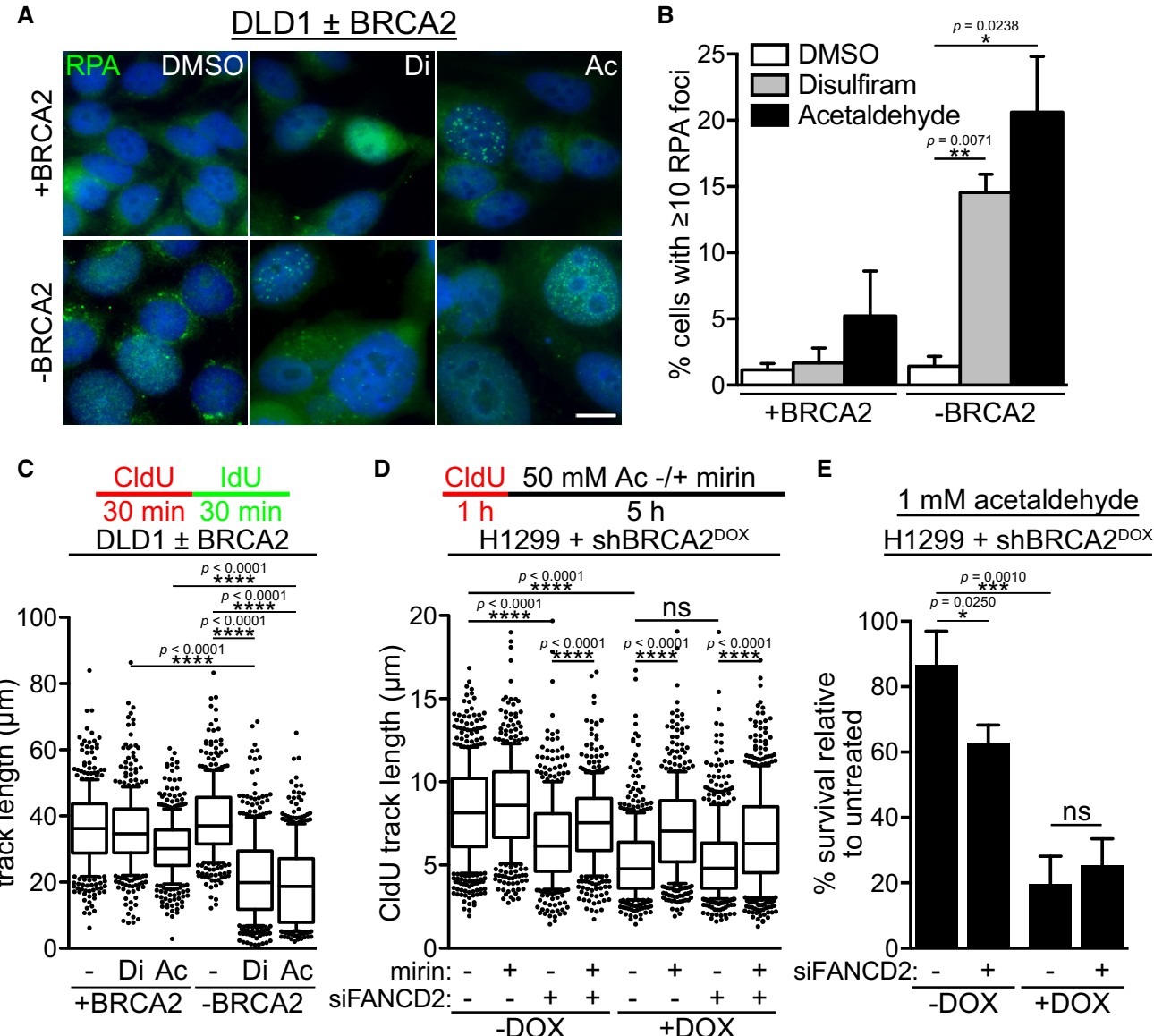

**Figure 3. Elevated replication stress and G2/M accumulation in BRCA2-deficient human cells treated with disulfiram or acetaldehyde.**

A Representative images of BRCA2-proficient (+BRCA2) and BRCA2-deficient (−BRCA2) human DLD1 cells incubated with 10 μM disulfiram or 4 mM acetaldehyde for 4 days prior to processing for immunofluorescence staining with anti-RPA antibody (green). DNA was counter-stained with DAPI (blue). Scale bar, 10 μm.

B Quantification of the frequency of BRCA2-proficient (+BRCA2) or BRCA2-deficient (−BRCA2) human DLD1 cells with 10 or more RPA foci in cells treated as in (A). Error bars represent SD of two independent experiments. *P*-values were calculated using an unpaired two-tailed *t*-test.

C BRCA2-proficient (+BRCA2) or BRCA2-deficient (−BRCA2) human DLD1 cells were treated with 2.5 μM disulfiram or 2 mM acetaldehyde overnight and processed for DNA fiber analysis, followed by quantification of the replication track length. Data were obtained from three independent experiments. Middle line represents median, and the box extends from the 25th to 75th percentiles. The whiskers mark the 10th and 90th percentiles. *P*-values were calculated using the Mann–Whitney test. -, DMSO; Di, disulfiram; Ac, acetaldehyde.

D H1299 cells expressing a DOX-inducible BRCA2 shRNA were grown in the presence or absence of DOX, with or without siRNA-mediated depletion of FANCD2. Processing for DNA fiber analysis was followed by quantification of CldU track length to measure replication fork stability under 50 mM acetaldehyde treatment, which blocks replication. Data were obtained from two independent experiments. Middle line represents median, and the box extends from the 25th to 75th percentiles. The whiskers mark the 10th and 90th percentiles. *P*-values were calculated using the Mann–Whitney test. DOX, doxycycline.

E H1299 cells treated as in (D) were grown in the presence of 1 mM acetaldehyde for clonogenic survival assays. Error bars represent SD of three independent experiments, each performed in triplicate. *P*-values were calculated using an unpaired two-tailed *t*-test. DOX, doxycycline.

resulted only in a small increase in the frequency of cells with 10 or more RPA foci. In order to directly evaluate the level of replication stress induced by acetaldehyde or disulfiram, we performed DNA fiber analyses, which allowed quantification of replication fork progression. We observed that replication track length was specifically and significantly reduced in the absence of BRCA2, upon treatment with either disulfiram or acetaldehyde (Fig 3C and Appendix Table S1).

Hydroxyurea-stalled replication forks require BRCA2 and FANCD2 for protection against MRE11-dependent nucleolytic degradation (Michl *et al*, 2016a). We therefore addressed whether acetaldehyde treatment can elicit a similar effect. Prolonged incubation of human H1299 cells with 50 mM acetaldehyde triggered a global replication arrest comparable to hydroxyurea (data not shown). Nascent DNA tracks were significantly shorter in FANCD2- or BRCA2-depleted cells (Fig 3D and Appendix Table S2). Importantly, concomitant inactivation of FANCD2 in BRCA2-depleted cells did not result in additional fork resection, implicating the two proteins in the same pathway of stalled fork protection in response to acetaldehyde treatment. Mirin inhibition of MRE11 nuclease activity rescued shortening of nascent DNA tracks in cells lacking BRCA2 and FANCD2. This identifies MRE11 as one nuclease implicated in fork degradation in this setting. Consistent with BRCA2 and FANCD2 promoting the same mechanism of replication fork stability in acetaldehyde-treated cells, inactivation of FANCD2 in BRCA2-depleted cells did not have an additional impact on cell survival in the presence of acetaldehyde (Fig 3E). Taken together, these results suggest a concerted action of FANCD2 and BRCA2 in the cellular response to acetaldehyde treatment.

## DNA damage accumulation and apoptosis in BRCA2-deficient cells treated with acetaldehyde and disulfiram

We next addressed whether the replication defects inflicted by acetaldehyde and disulfiram could trigger increased DNA damage in the context of BRCA2 deficiency. The levels of γH2AX foci, an important DSB marker, were initially used to evaluate DSB formation upon acetaldehyde accumulation (Fig 4A). This assay revealed a significant induction of γH2AX foci in BRCA2-deficient DLD1 cells upon disulfiram or acetaldehyde treatment relative to untreated cells (from 28.6% untreated cells with ten or more γH2AX foci to 43.7 and 66.7%, respectively). The frequency of cells with 10 or more γH2AX foci in untreated BRCA2-deficient DLD1 cells (28.6%) was elevated compared to wild-type counterparts (2.3%), reflecting increased levels of genomic instability upon loss of BRCA2.

In order to evaluate the full extent of DNA damage in response to acetaldehyde accumulation, we used metaphase chromosome spreads to directly visualize chromosome aberrations, including chromatid/chromosome breaks and genomic rearrangements (Fig EV4). This analysis demonstrated induction of chromosome aberrations by disulfiram or acetaldehyde, preferentially in the BRCA2-deficient cells (from an average of 5.9 per metaphase in untreated cells to 13.1 and 14.8, respectively; Fig 4B). It is noteworthy that the number of chromosomal aberrations evaluated with this assay represented an underestimation, since excessive DSBs prevent entry into mitosis and thus a large fraction of cells is excluded from this quantification. These data nevertheless clearly indicate that

disulfiram and acetaldehyde treatments inflict higher levels of DNA damage in BRCA2-deficient cells.

Next, we studied the impact of unrepaired DNA breaks on cell cycle progression using FACS analyses. Both disulfiram and acetaldehyde caused a significant accumulation of BRCA2-deficient cells in G2/M, relative to untreated controls (Fig 4C). Neither treatment affected significantly the proportion of BRCA2-proficient cells in G2/M.

In order to understand the causes of G2/M arrest elicited by disulfiram and acetaldehyde, we performed immunoblot analyses of key DNA damage response proteins, including checkpoint factors. Consistent with the IF results, we observed an increase in γH2AX levels specifically in BRCA2-deficient DLD1 cells upon treatment with either compound (Fig 4D). Furthermore, a clear induction of phospho-KAP1 and p53 expressions was observed specifically in BRCA2-deficient DLD1 cells upon disulfiram or acetaldehyde treatment, indicative of ATM-dependent DNA damage signaling (Guo *et al*, 2014). These data demonstrate a clear and specific activation of checkpoint signaling in BRCA2-deficient cells, consistent with the persistent DNA damage inflicted by disulfiram and acetaldehyde in BRCA2-deficient cells.

We furthermore monitored induction of apoptosis and how this was temporally related to checkpoint activation. Treatment with either disulfiram or acetaldehyde led to robust induction of cleaved PARP1, a marker for apoptosis, specifically in BRCA2-deleted cells (Fig 4E), suggesting that the toxicity of these compounds in due in part to apoptosis. We monitored both cleaved PARP1 and phospho-KAP1 levels in response to disulfiram or acetaldehyde treatments over a 5-day period. In BRCA2-deficient cells, phospho-KAP1 induction by disulfiram occurred after 1 day of treatment, preceding cleaved PARP1, induced after 2 days. A similar pattern was detected during acetaldehyde exposure, with phospho-KAP1 levels increasing after 2 days, prior to PARP1 cleavage during day three of treatment. Taken together, these data support the concept that toxic DNA damage is induced specifically and persists in BRCA2-deficient cells upon acetaldehyde build-up, with deleterious consequences for cell survival.

## HR abrogation inhibits proliferation of *Aldh2*$^{-/-}$ MEFs

Although disulfiram effectively inhibited ALDH activity in human cells, this did not exclude the possibility that the observed effects on cell viability could be due to secondary effects of the drug, beyond its role in acetaldehyde metabolism (Bruning & Kast, 2014). Therefore, we next sought to recapitulate the results obtained with chemical ALDH inhibition using genetic approaches. This was facilitated by the availability of immortalized MEFs carrying *Aldh2* gene deletion (*Aldh2*$^{-/-}$) and wild-type *Aldh2*$^{+/+}$ controls (Langevin *et al*, 2011). Consistent with previous data demonstrating that ALDH2 provides the major ALDH activity in MEFs (Garaycoechea *et al*, 2012), we detected a significant reduction in ALDH activity in *Aldh2*$^{-/-}$ MEFs using ALDEFLUOR™ assays (Appendix Fig S4A–C).

To investigate potential synthetic lethality between HR factors and ALDH2, we inhibited BRCA1, BRCA2, or RAD51 expressions in immortalized *Aldh2*$^{+/+}$ and *Aldh2*$^{-/-}$ MEFs and monitored the impact on cell proliferation. BRCA1 shRNA-mediated depletion,

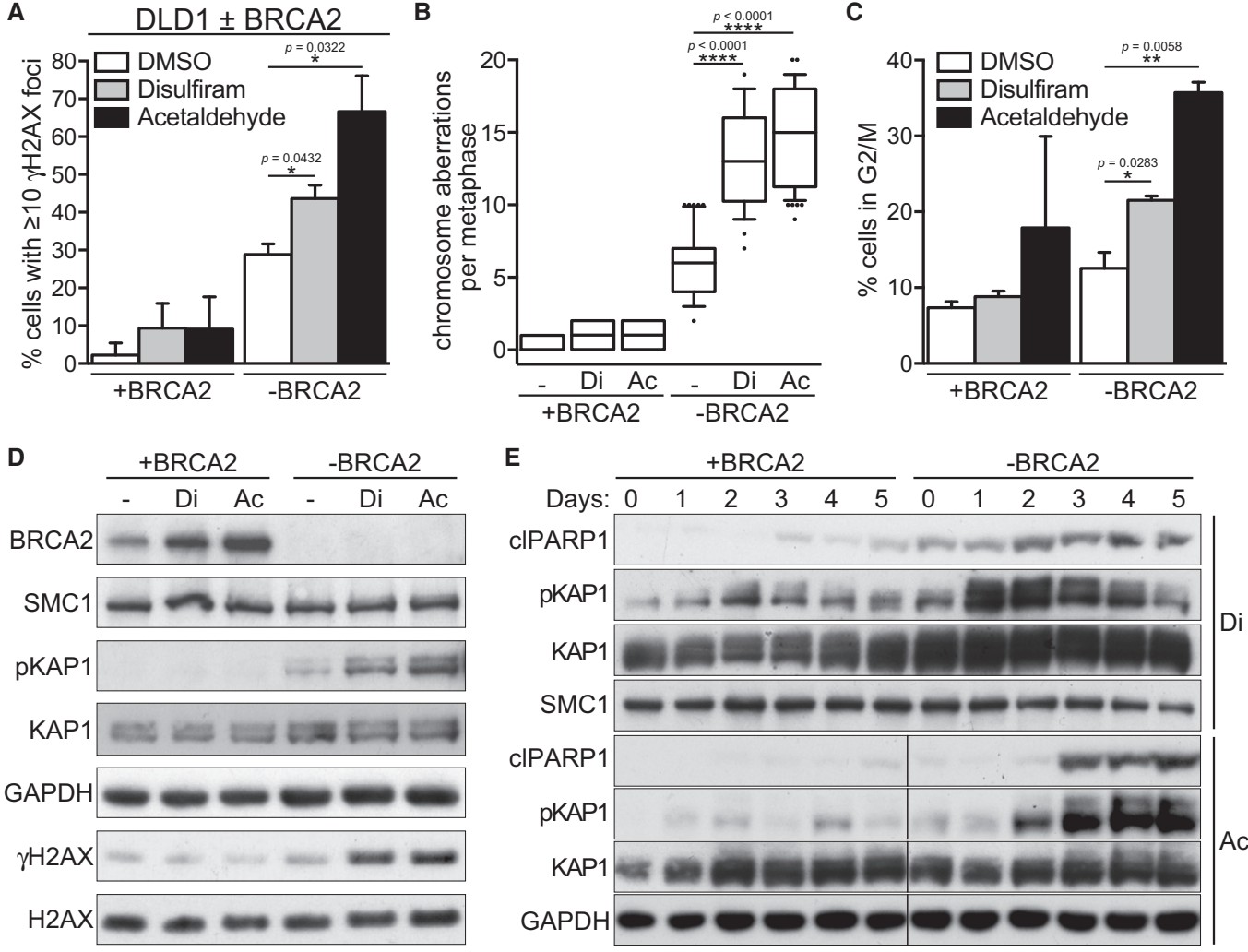

**Figure 4.  Disulfiram and acetaldehyde induce DSBs, chromosome aberrations, and checkpoint activation in BRCA2-deficient human cells.**

A  Quantification of the frequency of cells with 10 or more γH2AX foci in BRCA2-proficient (+BRCA2) or BRCA2-deficient (−BRCA2) human DLD1 cells incubated with 10 μM disulfiram or 4 mM acetaldehyde for 4 days. Error bars represent SD of two independent experiments. *P*-values were calculated using an unpaired two-tailed *t*-test.

B  Cells treated as in (A) were incubated overnight with colcemid and metaphase spreads were stained with Giemsa. To quantify chromosome aberrations, approximately 50 metaphases were analyzed for each sample. Middle line represents median, and the box extends from the 25th to 75th percentiles. The whiskers mark the 10th and 90th percentiles. *P*-values were calculated using an unpaired two-tailed *t*-test. -, DMSO; Di, disulfiram; Ac, acetaldehyde.

C  Cells treated as in (A) were processed for FACS analyses of DNA content. Quantification of the percentage of cells in G2/M is shown. Error bars represent SD of two independent experiments. *P*-values were calculated using an unpaired two-tailed *t*-test.

D  Whole-cell extracts prepared from BRCA2-proficient or BRCA2-deficient human DLD1 cells treated as in (A) were immunoblotted as indicated. SMC1, GAPDH, and H2AX were used as loading controls. -, DMSO; Di, disulfiram; Ac, acetaldehyde.

E  BRCA2-proficient (+BRCA2) and BRCA2-deficient (−BRCA2) human DLD1 cells were incubated with 10 μM disulfiram or 4 mM acetaldehyde. Whole-cell extracts prepared at the indicated time points during treatment were immunoblotted as shown. KAP1, SMC1, and GAPDH were used as loading controls. Di, disulfiram; Ac, acetaldehyde.

Source data are available online for this figure.

reflected in the reduced mRNA levels using quantitative PCR (qPCR) (Fig 5A), caused a significant decrease in the proliferation of $Aldh2^{-/-}$ MEFs (Fig 5B). *Brca2* gene deletion using CRISPR/Cas9 lentiviral system led to loss of BRCA2 expression (Fig 5C) and inhibited cell proliferation (Fig 5D). Importantly, while $Aldh2^{+/+}$, $Aldh2^{-/-}$, and $Brca2^{-/-}Aldh2^{+/+}$ single-cell clones grew normally, we could not establish $Brca2^{-/-}Aldh2^{-/-}$ single-cell clones, suggestive of a synthetic lethal interaction between the two genes.

Retroviral RAD51 shRNA infection of $Aldh2^{+/+}$ and $Aldh2^{-/-}$ MEFs led to efficient suppression of RAD51 expression, as assessed by immunoblotting (Fig 5E). While RAD51 depletion did not have a marked effect on $Aldh2^{+/+}$ MEFs, it substantially impaired the proliferative capacity of the $Aldh2^{-/-}$ MEFs (Fig 5F).

In similar experiments, we analyzed human fibroblasts carrying the naturally occurring rs671 *ALDH2* mutation (Hui *et al*, 2014). A single G to A nucleotide change causes the substitution of glutamate

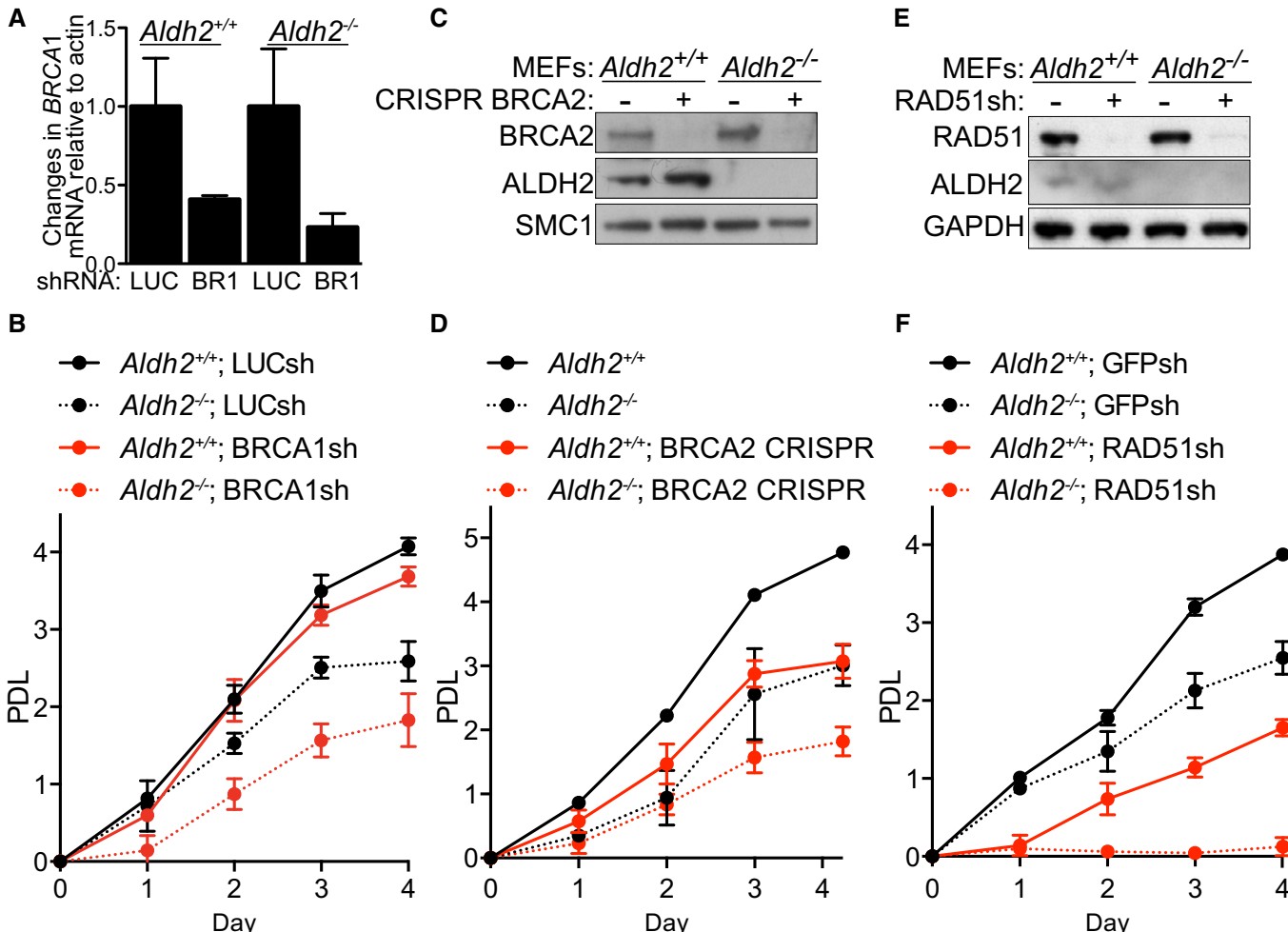

**Figure 5. Aldh2 gene deletion in MEFs is synthetic lethal with HR abrogation.**

A   Aldh2⁺/⁺ and Aldh2⁻/⁻ MEFs were infected with lentiviruses expressing luciferase control or BRCA1 shRNAs, followed by selection with puromycin for 72 h. mRNA was isolated and the level of Brca1 transcript was determined using quantitative PCR. Graphs are representative of two independent experiments, each performed in triplicate. Error bars represent SD of triplicate values obtained from a single experiment.

B   Proliferation rates of cells treated as in (A). Three days post-selection cells were plated in 96-well plates, and proliferation was determined using a resazurin-based assay at 24-h intervals for 4 days. Graphs are representative of two independent experiments, each performed in triplicate. Error bars represent SD of triplicate values obtained from a single experiment. PDL, population doubling.

C   Aldh2⁺/⁺ and Aldh2⁻/⁻ MEFs were infected with CRISPR lentiviruses expressing Cas9-P2A-Puro and gRNAs targeting mouse Brca2 or control vectors, followed by selection with puromycin 72 h. Cell extracts representative of the entire cell population were prepared and immunoblotted as indicated. SMC1 was used as a loading control.

D   Proliferation rates of cells treated as in (C). Graphs are representative of three independent experiments, each performed in triplicate. Error bars represent SD of triplicate values obtained from a single experiment.

E   Aldh2⁺/⁺ and Aldh2⁻/⁻ MEFs were infected with retroviruses expressing GFP control or RAD51 shRNAs, followed by selection with puromycin for 72 h. Cell extracts were prepared and immunoblotted as indicated. GAPDH was used as a loading control.

F   Proliferation rates of cells treated as in (E). Graphs are representative of three independent experiments, each performed in triplicate. Error bars represent SD of triplicate values obtained from a single experiment.

Source data are available online for this figure.

to lysine at position 487 (E487K). In homozygous carriers, this substitution is associated with the ethanol-induced flushing syndrome (Yoshida *et al*, 1984). Using cell lines derived from *ALDH2* mutant and wild-type individuals, we observed impaired proliferation specifically when BRCA1, BRCA2, or RAD51 were depleted in fibroblasts established from human patients homozygous for the *ALDH2* E487K mutation (Fig EV5). These results

corroborate our data obtained in mouse embryonic fibroblasts carrying *Aldh2* gene deletion. Given that approximately 560 million East Asians (8% of the world population) carry this mutant allele (Chen *et al*, 2014), our results gain a substantial translational relevance as they predict a low occurrence of BRCA2-mutated breast and ovarian cancers in this population. These results support the notion that genetic targeting of acetaldehyde metabolism in conjunction with

HR leads to growth arrest, highlighting the specific requirement for HR in counteracting the genotoxic effects of endogenous acetaldehyde accumulation.

## Acetaldehyde kills Brca1$^{-/-}$ and Brca2$^{-/-}$ mouse mammary tumor-derived cells and inhibits growth of BRCA2-deficient tumors

BRCA2-deficient mouse mammary tumor-derived cell lines recapitulate the responses of human breast cancers to a variety of drugs (Bouwman & Jonkers, 2014). We thus studied the impact of acetaldehyde on the viability of Brca2-deleted (KB2P3.4) mouse mammary tumor cell lines, relative to Brca2 wild-type controls (KB2P3.4R3; Fig 6A). Acetaldehyde treatment led to a specific reduction in the viability of BRCA2-deficient cells, an effect similar to that of olaparib. Of note, disulfiram was found to be particularly toxic to mouse cells and could not be used in similar experiments. Moreover, Brca1$^{-/-}$ mouse tumor cells also displayed acetaldehyde sensitivity relative to Brca1$^{+/+}$ control cells (Fig 6B). Survival assays confirmed the sensitivity of both Brca1- and Brca2-deleted mouse mammary tumor-derived cells lines to acetaldehyde (Appendix Fig S5A–D). Importantly, a related Brca1$^{-/-}$ mouse tumor cell line, which acquired olaparib resistance as a result of 53BP1 inactivation (Jaspers et al, 2013), also showed remarkable sensitivity to acetaldehyde in viability assays (Fig 6B). Acetaldehyde toxicity to olaparib-resistant Brca1$^{-/-}$, 53BP1-deficient cells is due to their inability to activate HR, reflected in the reduced levels of acetaldehyde-induced RAD51 foci (Fig 6C and D). As a control, olaparib induced RAD51 foci in Brca1$^{-/-}$, 53BP1-deficient cells at levels comparable to Brca1$^{+/+}$ control cells. Brca1$^{-/-}$ cells failed to assemble RAD51 foci in the presence of either olaparib or acetaldehyde.

We next investigated whether the cellular responses to acetaldehyde can be recapitulated in vivo using xenograft models. To determine whether acetaldehyde toxicity to olaparib-resistant cells can be recapitulated in vivo, we established tumor allografts from Brca1$^{+/+}$, Brca1$^{-/-}$, and PARP inhibitor-resistant Brca1$^{-/-}$, 53BP1-deficient mouse mammary tumor cells. Acetaldehyde affected significantly ($P < 0.001$) the growth of Brca1$^{-/-}$, but not Brca1$^{+/+}$ tumors ($P = 0.17$; Fig 7A and B). Importantly, while PARP inhibitor treatment did not lead to a significant reduction in tumor weight inhibition (TWI), acetaldehyde inhibited the growth of allografts derived from Brca1$^{-/-}$, 53BP1-deficient cells by 54% ($P < 0.001$; Fig 7C). Furthermore, tumors established from BRCA2-proficient (Fig 7D) and BRCA2-deficient (Fig 7E) human DLD1 cells in nude mice were treated with acetaldehyde administered intravenously. Acetaldehyde had no significant effect ($P = 0.15$) on the growth of BRCA2-proficient tumors, as illustrated by of 12% TWI at the nadir of the effect (Fig 7D). In contrast, acetaldehyde effectively suppressed the growth of BRCA2-deficient tumors ($P < 0.001$), for which TWI assessed under similar conditions was 56% (Fig 7E) and produced stabilization of disease for the duration of treatment.

Next, we examined potential disulfiram toxicity against PDTCs, which represent a valuable resource for pre-clinical drug testing (Bruna et al, 2016). These short-term, ex vivo cell cultures established from patient-derived tumor xenografts (PDTXs) recapitulate tumor heterogeneity and response to various cancer drugs used in the clinic. PDTCs derived from three PDTX models were incubated for 6 days with disulfiram and its effect on cell viability was expressed relative to control DMSO treatment (Fig 7F). We observed that PDTCs from a tumor with no known BRCA1 alteration (AB521; http://caldaslab.cruk.cam.ac.uk/bcape/) did not respond to the drug. In contrast, PDTCs from two tumors carrying either BRCA1 promoter methylation (STG201) or BRCA1 germ line truncation (VHIO179) responded to disulfiram. Importantly, VHI179 carries a MAD2L2 inactivating mutation, which makes it resistant to olaparib (Bruna et al, 2016). PDTCs established from this tumor showed significantly reduced viability upon treatment with 1.25 μM disulfiram (Fig 7F). These results demonstrate that our conclusions based on cellular models can be translated in breast cancer patient-derived models. Most importantly, elevated acetaldehyde levels could be beneficial to the treatment of BRCA-deficient tumors in the clinic, including those that acquired PARP inhibitor resistance.

## Discussion

Acetaldehyde is generated both endogenously, as a product of cellular metabolism, and in response to exogenous factors, such as alcohol consumption. ALDHs play key roles in the detoxification of intracellular acetaldehyde pools. As a highly reactive molecule, acetaldehyde inflicts a broad spectrum of DNA damage, including ICLs (Stein et al, 2006; Voulgaridou et al, 2011). Although a large body of evidence supports the concept that acetaldehyde can induce DNA damage in vivo (Brooks & Theruvathu, 2005; Seitz & Stickel, 2007; Brooks & Zakhari, 2014), it remained unclear until recently how this damage is repaired. Recent work reported that mice deficient in both FA DNA repair pathway (Fancd2$^{-/-}$) and acetaldehyde catabolism (Aldh2$^{-/-}$) showed developmental aberrations upon exposure to ethanol and increased leukemia occurrence (Langevin et al, 2011). Aldh2$^{-/-}$Fancd2$^{-/-}$ double knockout mice that did not develop leukemia suffered bone marrow failure linked with DNA damage accumulation in the hematopoietic stem and progenitor cells (Garaycoechea et al, 2012). These results demonstrated for the first time the genotoxic effects of endogenous acetaldehyde accumulation in vivo and the key role of FANCD2 in counteracting them.

Hypersensitivity of Fancd2$^{-/-}$ cells to exogenous acetaldehyde (Langevin et al, 2011) and the extensive crosstalk between HR and FA factors in DNA replication and damage repair prompted us to investigate the potential HR role in the response to acetaldehyde. We demonstrate here that acetaldehyde hypersensitivity is also a feature of HR-compromised cells, lacking BRCA1, BRCA2, or RAD51. These cells cannot effectively replicate DNA or accurately repair toxic DSBs, leading to the accumulation of DNA damage, checkpoint activation, cell cycle arrest, and apoptosis.

Upon hydroxyurea-induced replication arrest, BRCA1, BRCA2, and FANCD2 are required for protection of stalled replication forks against MRE11-dependent nucleolytic degradation (Schlacher et al, 2011, 2012). Recent studies have shown that FANCD2 and BRCA1/2 promote independent mechanisms of fork protection in this setting (Kais et al, 2016; Michl et al, 2016a). Our current work demonstrates that, in contrast to hydroxyurea treatment, BRCA2 and FANCD2 act in the same pathway of stalled replication fork protection upon acetaldehyde-induced replication arrest. Acetaldehyde sensitivity of BRCA2-deficient cells is not further increased by FANCD2 inactivation, which further supports the epistatic

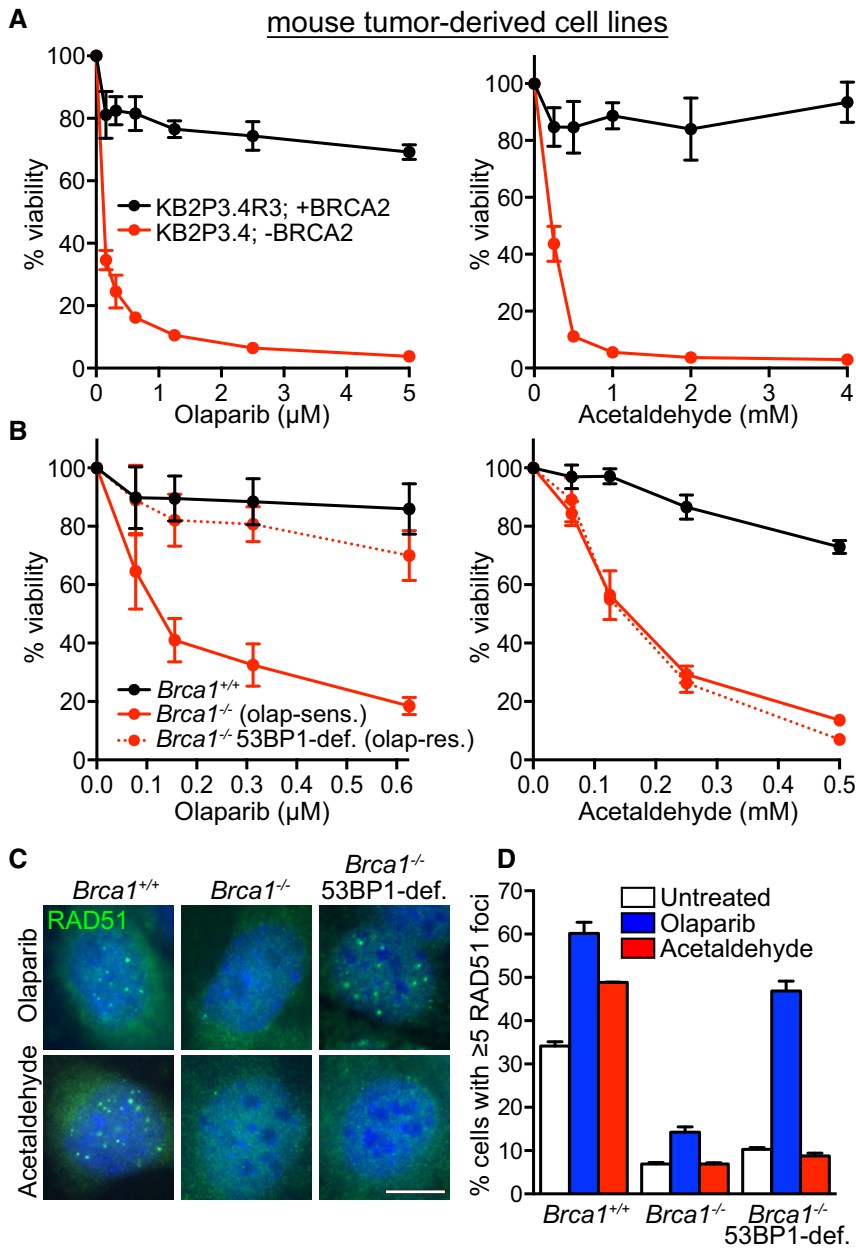

**Figure 6.  Acetaldehyde toxicity to BRCA2- and BRCA1-deficient mouse mammary tumor cell lines, including those that acquired olaparib resistance.**

A    Dose-dependent viability assays of mouse mammary tumor-derived *Brca2*$^{-/-}$ and control cell lines treated with olaparib or acetaldehyde at the indicated concentrations for 6 days. Graphs are representative of three independent experiments, each performed in triplicate. Error bars represent SD of triplicate values obtained from a single experiment. KB2P3.4R3 is a BRCA2-proficient tumor-derived cell line. KB2P3.4 is a BRCA2-deficient tumor-derived cell line.

B    Dose-dependent viability assays of *Brca1*$^{+/+}$ and *Brca1*$^{-/-}$ mouse mammary tumor-derived cell lines treated with olaparib or acetaldehyde at the indicated concentrations for 6 days. Graphs are representative of two independent experiments, each performed in triplicate. Error bars represent SD of triplicate values obtained from a single experiment. *Brca1*$^{+/+}$, BRCA1-proficient mouse tumor-derived cell line; *Brca1*$^{-/-}$ (olap-sens.), BRCA1-deficient, olaparib-sensitive mouse tumor-derived cell line; *Brca1*$^{-/-}$ 53BP1-def. (olap-res.), BRCA1- and 53BP1-deficient, olaparib-resistant mouse tumor-derived cell line.

C    *Brca1*$^{+/+}$ and *Brca1*$^{-/-}$ mouse mammary tumor-derived cell lines as in (B) were treated with 0.5 μM olaparib or 2 mM acetaldehyde for 48 h before processing for immunofluorescence staining with anti-RAD51 antibody (green). DNA was counter-stained with DAPI (blue). Scale bar, 5 μm.

D    Quantification of the frequency of cells with 5 or more RAD51 foci in cells treated as in (C). More than 200 nuclei were analyzed for each sample. Graphs are representative of two independent experiments. Error bars represent SD.

interaction between BRCA2 and FANCD2 in the response to acetaldehyde. These findings are consistent with FA and HR acting in the same pathway of ICL repair (Michl *et al*, 2016b).

We moreover found that disulfiram, an ALDH1A1 and ALDH2 inhibitor used in the clinic as an alcohol-aversive agent (Koppaka *et al*, 2012), reduces the viability of BRCA1/2-deficient cell lines

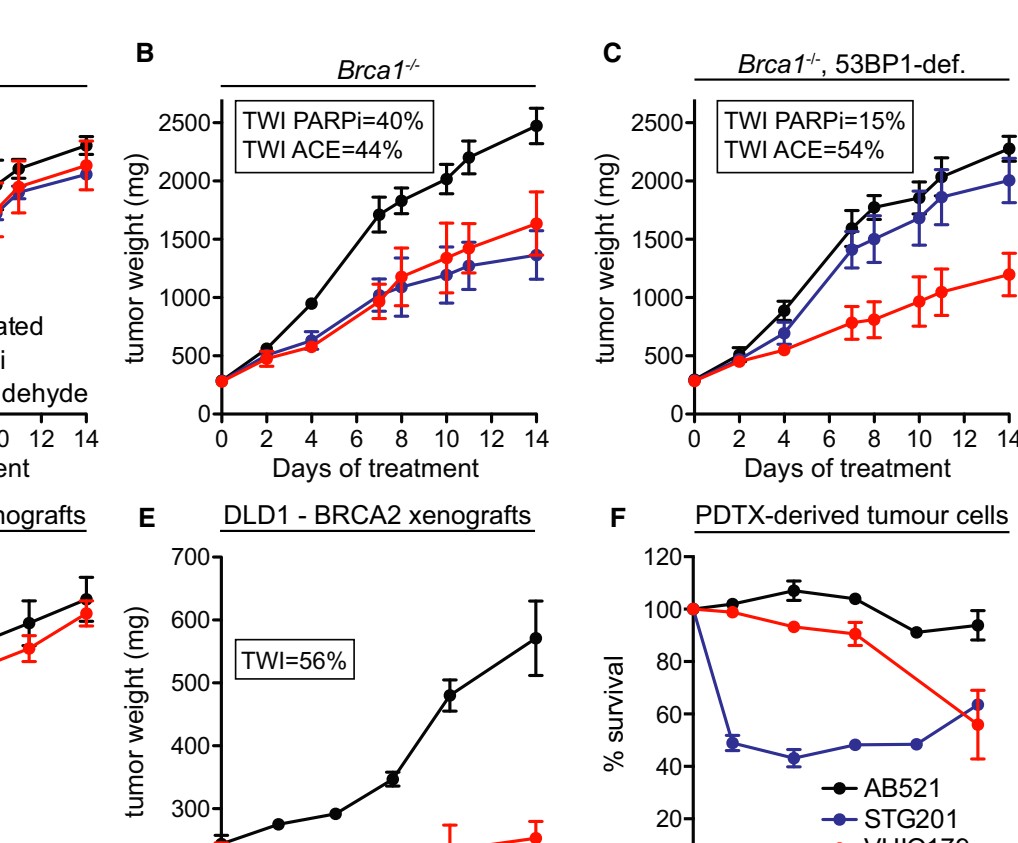

**Figure 7. Acetaldehyde treatment inhibits growth of BRCA1/2-deficient tumors in mouse models.**

A–C  Mice were injected with $Brca1^{+/+}$ (A), $Brca1^{-/-}$ (B), and $Brca1^{-/-}$, 53BP1-deficient (C) mouse mammary tumor cells. Treatment of tumor-bearing mice with 80 mg/kg acetaldehyde administered intravenously (i.v.) and 0.33 mg/kg PARP inhibitor Biomarine administered orally was initiated when a tumor mass of 250 mg was detected for each tumor. Tumor weight was assessed on the indicated days after initiation of the treatment. Each experimental group included five mice. Tumor weight inhibition (TWI) was calculated at the nadir of the effect (9 and 10 days of treatment for BRCA2-proficient and -deficient tumors, respectively). Error bars represent SD ($n = 5$). $P$-values for acetaldehyde versus untreated were calculated using an unpaired two-tailed $t$-test: (A), $P = 0.17$; (B), $P = 0.0000047$; (C), $P = 0.0000039$. PARPi, PARP inhibitor; ACE, acetaldehyde.

D, E  Mice were injected intramuscularly with $5 \times 10^6$ human DLD1 cells, BRCA2-proficient (D) or BRCA2-deficient (E). Error bars represent SD ($n = 5$). $P$-values for acetaldehyde versus untreated were calculated using an unpaired two-tailed $t$-test: (D), $P = 0.15$; (E), $P = 0.0000043$.

F  PDTCs derived from breast cancer samples as previously described (Bruna et al, 2016) were treated with disulfiram at the indicated doses. Cell survival is represented relative to DMSO control. AB521, ER-negative, no known $BRCA1$ alteration; STG201, $BRCA1$ promoter methylation and loss of $BRCA1$ expression; VHIO179, $BRCA1$ germ line mutation and $MAD2L2$ inactivating mutation (olaparib-resistant); http://caldaslab.cruk.cam.ac.uk/bcape/. Error bars represent SEM of triplicate values obtained from a single experiment.

and thus recapitulates the effect of exogenous acetaldehyde treatment. Similarly to acetaldehyde, disulfiram caused elevated levels of DNA damage, replication stress, and checkpoint activation specifically in cells lacking BRCA2. Our data support the notion that disulfiram-mediated endogenous acetaldehyde accumulation causes replication-associated DNA damage, leading to selective killing of BRCA2-deficient cells.

However, previous studies have attributed acetaldehyde metabolism-independent functions to disulfiram (Koppaka et al, 2012; Bruning & Kast, 2014). For example, reduced survival of prostate cancer-derived cells in the presence of disulfiram is thought to be due to the DNA-demethylating activity of the drug (Lin et al, 2011). Moreover, the copper-chelating capacity of disulfiram may cause proteasome inhibition and subsequent apoptosis in ovarian tumors

(Papaioannou et al, 2014). Definitive evidence supporting that disulfiram toxicity to BRCA1/2-deficient cells is due to its inhibitory effect on acetaldehyde metabolism comes from our genetic studies using $Aldh2^{+/+}$ and $Aldh2^{-/-}$ MEFs. Our ALDEFLUOR™ analysis demonstrated, consistent with previous results, that ALDH2 accounts for the majority of the ALDH activity in MEFs (Garaycoechea et al, 2012). Abrogation of HR pathway substantially reduced proliferation of cells harboring $Aldh2$ gene deletion, clearly pointing out to deregulated ALDH activity as the most probable mechanism underlying the sensitivity of HR-deficient cells to disulfiram. These data demonstrate that either functional acetaldehyde metabolism or functional HR repair is generally sufficient to overcome the genotoxic assault caused by elevated acetaldehyde levels. However, when both pathways are abrogated, the inability to metabolize acetaldehyde to less

toxic substrates, combined with ineffective repair of the acetaldehyde-induced DNA damage, becomes lethal to cells.

### Clinical implications

Mutations in *BRCA1* and *BRCA2* genes are associated with elevated risk of breast and ovarian cancer (Roy *et al*, 2012). A major barrier to initially effective therapies is the development of tumor cell resistance, which has been documented clinically for both platinum drugs and PARP inhibitors, the most promising approaches for treating HR-associated cancers thus far described (Ang *et al*, 2013; Ledermann *et al*, 2014). It is, therefore, crucial that effective, well-tolerated, and selective therapies for the targeting of HR-deficient tumor cells are rapidly developed.

The work presented here demonstrates that exogenous acetaldehyde effectively and selectively reduced the survival of several BRCA1/2-deficient mouse mammary tumor-derived cell lines, some of which recapitulate features of human breast tumors (Bouwman & Jonkers, 2014). Moreover, olaparib-resistant BRCA1-deficient mouse cell lines retained acetaldehyde sensitivity, highlighting the potential of drugs that interfere with acetaldehyde catabolism to be therapeutically relevant to patients whose tumors do not respond or have developed resistance to PARP inhibition.

Further supporting the clinical relevance of our findings, acetaldehyde treatment specifically inhibited growth of BRCA2-deficient human tumors in xenograft models and PARP inhibitor-resistant BRCA1-deficient allografted mammary tumors. Most importantly, we demonstrate disulfiram that disulfiram is specifically toxic to *BRCA1*-defective PDTCs. These are short-term cultures established from human breast tumors which recapitulate tumor heterogeneity and responses to therapy in culture (Bruna *et al*, 2016). BRCA1-deficient, but not BRCA1-proficient PDTCs were sensitive to disulfiram, thus confirming our model that HR-deficient cells and tumors can be effectively eliminated using this drug. As discussed above, our data support the view that acetaldehyde inflicts DNA damage, particularly in the form of toxic replication-associated DSBs, which cannot be accurately repaired in the absence of HR, thus explaining the exquisite vulnerability of BRCA1/2-deficient cells and tumors to acetaldehyde. While converting the vulnerability of BRCA1/2-deficient tumors to acetaldehyde into a therapeutic strategy may not be possible due to known genotoxic effects of this compound, exploiting natural acetaldehyde production seems a feasible approach. Disulfiram is highly toxic to human cells lacking BRCA1 or BRCA2 activity, due to its ability to increase endogenous acetaldehyde levels, which culminates in acetaldehyde-induced DSBs and cell death. As disulfiram is in widespread clinical use, its safety and tolerability are well established and it could be rapidly reassigned for treatment of cancers associated with BRCA1/2 mutations.

# Materials and Methods

### Cell lines and growth conditions

Human non-small cell lung carcinoma H1299 cells (American Type Culture Collection) and colorectal adenocarcinoma DLD1 cells, parental and *BRCA2*-mutated (Horizon Discovery; Zimmer *et al*, 2016), were cultivated in monolayers in DMEM medium (Sigma-Aldrich) supplemented with 10% fetal bovine serum (Life Technologies) and 1% penicillin/streptomycin (Sigma-Aldrich). H1299 cells expressing a doxycycline (DOX)-inducible BRCA1 or BRCA2 shRNAs were established using the "all-in-one" system (Wiederschain *et al*, 2009). The shRNAs targeting human BRCA1 (AGT ATG CAA ACA GCT ATA AT) or BRCA2 (GGG AAA CAC TCA GAT TAA A (Zimmer *et al*, 2016)) were cloned into pLKO$^{TetOn}$ and constructs were introduced into H1299 cells using lentiviral infection. Single-cell clones showed efficient BRCA1 or BRCA2 knockdown after 3 days in the presence of 10 μg/ml DOX in DMEM medium supplemented with 10% tetracycline-free fetal bovine serum (Clontech).

Immortalized $Aldh2^{+/+}$ and $Aldh2^{-/-}$ MEFs (a gift from Dr. K.J. Patel, LMB, Cambridge) and human fibroblasts carrying either WT or E487K homozygous *ALDH2* mutation were cultured at 37°C, 5% CO$_2$ and 3% oxygen in DMEM (Sigma-Aldrich) supplemented with 10% fetal bovine serum (Life Technologies), 1% penicillin/streptomycin (Sigma-Aldrich). Primary $Brca1^{F/-}$ primary MEFs (Bouwman *et al*, 2010; Carlos *et al*, 2013) were isolated from day 13.5 embryos as previously described (Blasco *et al*, 1997), immortalized by expression of p53 shRNA, and cultivated in a low-oxygen (3%) incubator. The mouse mammary tumor cell lines KP3.33 (BRCA1-proficient, control), KB1PM5 (BRCA1-deficient, PARP inhibitor-sensitive) and KB1PM5 [BRCA1-deficient, PARP inhibitor-resistant (Jaspers *et al*, 2013)] as well as $Brca2^{-/-}$ (KB2P1.21, BRCA2-deficient), $Brca2^{-/-}$+B2iBAC (KB2P1.21R2, BRCA2-proficient), $Brca2^{-/-}$ (KB2P3.4, BRCA2-deficient), $Brca2^{-/-}$+B2iBAC (KB2P3.4R3, BRCA2-proficient; Evers *et al*, 2010) were cultured at 37°C, 5% CO$_2$, and 3% oxygen in complete medium [DMEM/F-12, (Life Technologies) supplemented with 10% fetal bovine serum (Life Technologies), 1% penicillin/streptomycin (Sigma-Aldrich), 5 μg/ml insulin (Sigma-Aldrich), 5 ng/ml epidermal growth factor (Life Technologies), and 5 ng/ml cholera toxin (Gentaur)]. All cell lines used were routinely tested for mycoplasma. Acetaldehyde (Sigma-Aldrich), disulfiram (Prestwick/Selleckchem), cisplatin (Sigma-Aldrich), and olaparib (Selleckchem) were added to the media in the concentrations indicated. For mitotic arrest, cells were treated with 0.2 μg/ml KaryoMAX® colcemid (Life Technologies) overnight.

### RNAi

Cells were transfected using Dharmafect 1 (Dharmacon Research). Briefly, $0.8 \times 10^6$ to $1.5 \times 10^6$ cells were transfected with 40 nM siRNA per plate by reverse transfection in 10-cm plates. After 48-h incubation, depletion was evident as determined by immunoblotting. RAD51 esiRNA was obtained from Sigma-Aldrich (EHU045521), BRCA2 and FANCD2 siGENOME SMARTpool from Dharmacon (M-003462-01 and M-016376-02, respectively), BRCA1 and AllStars negative control siRNAs were obtained from Qiagen (S102654575 and 1027281, respectively).

### MEF retroviral transduction

Retroviral transduction of cultured MEFs was performed as previously described (Palmero & Serrano, 2001). In brief, HEK293T packaging cells were grown to 70% confluency and transfected with pCL-Eco helper vector together with either pBabe plus retroviral vector encoding "Hit-and-run" Cre recombinase (Silver &

Livingston, 2001) or pRetroSuper retroviral constructs encoding either a mouse 53BP1 shRNA (GCT ATT GTG GAG ATT GTG TTT), mouse BRCA1 shRNA (GCC TCA CTT TAA CTG ACG CAA T, used as a 1:1 mix with lentiviral-expressed BRCA1 shRNA, see sequence below), mouse RAD51 shRNA (AGA ATG TCT CAC AAA TAA), or control GFP shRNA (GCT GAC CCT GAA GTT CAT CTT) using a standard calcium phosphate protocol. The medium was replaced 24 h after transfection. Recipient MEFs were plated and infected 24 h later with the retroviral supernatants produced by the HEK293T cells. Additional infections were performed after 24 and 32 h. Twenty-four hours following the last infection, cells were incubated in fresh medium containing 3 μg/ml puromycin for 72 h. Exponentially growing MEFs were collected and processed for immunoblotting, cell viability, and proliferation assays.

### MEF lentiviral transduction and CRISPR/Cas9-mediated *Brca2* deletion

For lentiviral transduction of cultured MEFs (Dull *et al*, 1998), HEK293T packaging cells were grown to 70% confluency and transfected with Gag-Pol, Res-Rev, and VSV-G packaging vectors together with either the lentiCRISPR vector expressing Cas9-P2A-Puro (Cong *et al*, 2013) and mouse *Brca2* gRNA (GCC CTT ACG CCT GAC TCC GT) or the shRNA-containing pLKO.1 constructs: control luciferase shRNA (CGC TGA GTA CTT CGA AAT GTC) and BRCA1 shRNA (Sigma-Aldrich; TRCN306222; GTG CTT CCA CAC CCT ACT TAC, used as a 1:1 mix with retroviral-expressed BRCA1 shRNA). The medium was replaced 24 h after transfection. Recipient MEFs were plated and infected twice at 12-h intervals with the lentiviral supernatants produced by the HEK293T cells. 24 h after the last infection, cells were washed and incubated in fresh medium containing 3 μg/ml puromycin for 3–5 days. MEFs were processed as described for the retroviral transduction.

### Real-time quantitative PCR (qPCR)

Real-time RT–PCR using the PowerSYBR® Green Cells-to-CT™ kit (Ambion) was performed after puromycin selection in order to determine *Brca1* mRNA levels. Reverse transcriptase reactions were performed in a Veriti® 96-Well Thermal Cycler (Applied Biosystems, Life Technologies Corporation). PCRs were performed using a StepOnePlus™ Real-Time PCR System (Applied Biosystems, Life Technologies Corporation). The primers used to amplify mouse *Brca1* mRNA: ATG CTC TGG CAG CAT GTT CT and CAC TCT GCG AGC AGT CTT CA. Primers for mouse β-actin (GCT CTG GCT CCT AGC ACC AT and CCA CCG ATC CAC ACA GAG TAC) were used as an endogenous control. All qPCRs were performed in triplicate.

### Cell viability and proliferation assays

Cells were plated at densities varying between 1,000 and 3,000 cells per well in 96-well plates, and drugs were added at the indicated concentrations. Cell viability was determined by incubating cells with medium containing 10 μg/ml of resazurin for 2 h. Fluorescence was measured at 590 nm using a plate reader (POLARstar, Omega). To determine population doublings, resazurin-based read-outs of cell viability were taken after cells had adhered (day 0) and

at 24-h intervals for 4 days. Cell viability was expressed relative to untreated cells of the same cell line, thus accounting for any differences in viability caused by HR deficiency.

### Clonogenic assays

Cells were plated at densities between 200 and 1,000 cells per well in 6-well plates, and drug treatment was initiated after cells had adhered. Following 24-h incubation with the drug, fresh media without the drug were added for 10–14 days. Colonies were stained with 5 mg/ml crystal violet (Sigma-Aldrich) in 50% methanol, 20% ethanol in dH$_2$O. Cell survival was expressed relative to untreated cells of the same cell line, thus accounting for any differences in viability caused by HR deficiency.

### Immunoblotting

To prepare whole-cell extracts, cells were washed once in PBS, harvested by trypsinization, washed in cold PBS, and resuspended in SDS–PAGE loading buffer. Samples were sonicated using a probe sonicator and heated at 70°C for 10 min. Equal amounts of protein (50–100 μg) were analyzed by gel electrophoresis followed by Western blotting. NuPAGE-Novex 10% Bis–Tris and NuPAGE-Novex 3–8% Tris–Acetate gels (Life Technologies) were run according to manufacturer's instructions.

### Immunofluorescence

Cells cultured on coverslips were washed in PBS, swollen in hypotonic solution (85.5 mM NaCl and 5 mM MgCl$_2$) for 5 min and stained as previously described (Zimmer *et al*, 2016). In brief, cells were fixed with 4% paraformaldehyde for 10 min at room temperature (and with 100% ice-cold methanol for RPA) and permeabilized by adding 0.03% SDS to the fixative. After blocking with blocking buffer (1% goat serum, 0.3% BSA, 0.005% Triton X-100 in PBS), cells were incubated with primary antibody diluted in blocking buffer overnight at room temperature. Then, they were washed again and incubated with fluorochrome-conjugated secondary antibodies for 1 h at room temperature. Dried coverslips were mounted on microscope slides using the ProLong Antifade kit (Life Technologies) supplemented with 2 μg/ml DAPI. Samples were viewed with a Leica DMI6000B inverted microscope and fluorescence imaging workstation equipped with a HCX PL APO 100×/1.4–0.7 oil objective. Images were acquired using a Leica DFC350 FX R2 digital camera and LAS-AF software (Leica).

### ALDEFLUOR™ assay

Quantification of ALDH activity was performed using an ALDE-FLUOR™ Kit (STEMCELL Technologies), according to manufacturer's instructions. Cells were washed once in PBS, harvested by trypsinization, washed with PBS again, and resuspended in 1 ml ALDEFLUOR™ assay buffer. Cells were counted and the concentration adjusted to 1 × 10$^6$ cells per 1 ml assay buffer. 5 μl of activated ALDEFLUOR™ reagent was then added to 1 ml of cells, and 500 μl of the mix was immediately transferred to a "control" tube containing 5 μl DEAB, a generic ALDH inhibitor used as control for background fluorescence. Samples were incubated at 37°C for 60 min,

spun at 250 RCF for 5 min in a bench-top centrifuge, and resuspended in 500 μl ALDEFLUOR™ assay buffer prior to FACS analysis. One hundred thousand cells were analyzed by flow cytometry (Becton Dickinson). Data were acquired and analyzed using CellQuest software (Becton Dickinson).

### DNA fiber assays

For fork progression analysis, the cells were treated with disulfiram (2.5 μM) and acetaldehyde (2 mM) overnight. After the treatment, DNA of replicating cells was labeled with 25 μM CldU for 30 min, followed by labeling with 250 μM IdU for 30 min. For fork protection experiments, the cells were first labeled with 25 μM CldU for 60 min, followed by acetaldehyde (50 mM) with or without mirin (50 μM) for 5 h. The reaction was terminated by addition of ice-cold PBS. Cells were harvested and resuspended in cold PBS at $5 \times 10^5$ cells/ml. Next, 7 μl of lysis buffer (200 mM Tris–HCl pH 7.4, 50 mM EDTA, 0.5% SDS) was mixed with 2 μl of cell suspension on a microscopy slide and incubated horizontally for 9 min at RT. The DNA was spread by tilting the slide manually at an angle of 30°–45°. The air-dried DNA was fixed in methanol/acetic acid (3:1) for 10 min. Slides were then rehydrated in PBS twice for 3 min, and the DNA was denatured in 2.5 M HCl for 1 h at RT. The slides were washed several times in PBS until a pH of 7–7.5 was reached, followed by incubation in blocking solution (2% BSA, 0.1% Tween 20, PBS; 0.22 μm filtered) for 40 min at RT and in primary antibodies (rat anti-CldU, 1:500 (Abcam) and mouse anti-IdU, 1:100 (Beckton Dickinson)) for 2.5 h at RT. After five washes in PBS–Tween (0.2% Tween 20 in PBS) for 3 min and one short wash in blocking solution, the slides were incubated with the secondary antibodies (anti-rat Cy3 and anti-mouse FITC) for 1 h at RT. Subsequently, they were washed as before, air-dried, and mounted in Antifade Gold (Life Technologies). Images were acquired as described for IF and analyzed using ImageJ software (National Healthcare Institute, USA).

### FACS analysis

Cells were washed once in PBS, harvested by trypsinization, washed in cold PBS, and fixed in ice-cold 70% ethanol for at least 12 h at 4°C. Following fixation, cells were washed twice in cold PBS and incubated with 20 μg/ml propidium iodide (Sigma-Aldrich) and 10 μg/ml RNase A (Sigma-Aldrich) in PBS for 30 min at 37°C. A minimum of 10,000 cells were analyzed by flow cytometry (Becton Dickinson). Data were processed using CellQuest and ModFit LT software.

### Preparation of metaphase chromosome spreads

Cells synchronized in mitosis via overnight incubation with 0.1 μg/ml KaryoMAX® colcemid (Life Technologies) were collected by mitotic shake-off and swollen in hypotonic buffer (0.03 M sodium citrate) at 37°C for 25 min. Cells were fixed in a freshly prepared 3:1 mix of methanol:glacial acetic acid, and nuclear preparations were dropped onto slides pre-soaked in 45% acetic acid prior to being allowed to dry overnight. The following day, mitotic chromosomes were stained using Giemsa (VWR) and viewed as described for IF.

### *In vivo* drug experiments

CD-1 male nude (nu/nu) mice (6 weeks old and weighing 26–28 g) and FVB female mice (5 weeks old and weighing 24–26 g) were purchased from Charles River Laboratories (Calco, Italy). All animal procedures were in compliance with the national and international directives (D.L. March 4, 2014, no. 26; directive 2010/63/EU of the European Parliament and of the council; Guide for the Care and Use of Laboratory Animals, United States National Research Council, 2011).

Nude mice were injected intramuscularly into the hind leg muscles with $5 \times 10^6$ DLD1 BRCA2-proficient or -deficient cells per mouse. When a tumor mass of about 250 mg was evident in BRCA2-proficient (4 days after cell injection) and BRCA2-deficient (6 days after cell injection) xenografts, the treatment was initiated. FVB mice were injected intramuscularly into the hind leg muscles with $4 \times 10^6$ cells per mouse of $Brca1^{+/+}$, $Brca1^{-/-}$ and PARP inhibitor-resistant $Brca1^{-/-}$, 53BP1-deficient mouse mammary tumor cells. When a tumor mass of about 250 mg was evident (3 days after cell injection), the treatment was initiated. Each experimental group included five mice.

Based on previously established maximum tolerated dose (Isse *et al*, 2005), acetaldehyde (80 mg/kg) was administered intravenously for ten consecutive days. PARP inhibitor Biomarine (0.33 mg/kg) was administered orally by gavage for five consecutive days for two cycles of treatment. Tumors were measured in two dimensions using a caliper at the time points indicated, and tumor weight was calculated using the formula $a \times b^2/2$, where a and b are the long and short sizes of the tumor, respectively. Each experimental group included five mice. Therapeutic efficacy was measured as percent tumor weight inhibition (TWI%), calculated as [1−(mean tumor weight of treated mice/mean tumor weight of controls)] × 100.

### *Ex vivo* drug experiments

The *ex vivo* drug treatment protocol was adapted from (Bruna *et al*, 2016). Briefly, frozen patient-derived tumor xenografts (PDTXs) were thawed and dissociated on the GentleMACS Dissociator (Miltenyi Biotec, Cat ID 130-093-235) using the Tumor Dissociation Kit, human (Miltenyi Biotec, Cat ID 130-095-929) and preset protocol h_tumour_01. Single cells were plated at ~40,000 cells/ml in 96-well plates and dosed 72 h after plating. Cell Titer Glo 3D was added to the cells 6 days after dosing. Plates were read on the Pherastar plate reader using the Luminescence module.

### Antibodies

The following antibodies were used for immunoblotting: rabbit polyclonal antisera raised against H2AX (DR1016, Calbiochem), phosphorylated KAP1 (A300-767A, Bethyl Laboratories), KAP1 (A300-274A, Bethyl Laboratories), cleaved PARP1 Asp214 (9541, Cell Signaling), RAD51 (H92, Santa Cruz Biotechnology), ALDH2 (ab108306, Abcam), FANCD2 (NB100-182, Novus Biologicals), and SMC1 (BL308, Bethyl Laboratories); mouse monoclonal antibodies raised against BRCA1 (OP92, Calbiochem), BRCA2 (OP95, Calbiochem), GAPDH (6C5, Novus Biologicals), and phosphorylated histone H2AX Ser139 (clone JBW301, Merck Millipore); a sheep

**The paper explained**

**Problem**

Heterozygous *BRCA1* and *BRCA2* mutations increase the risk of breast and ovarian cancers. At the molecular level, BRCA1 and BRCA2 proteins play essential roles in DNA replication and double-strand break repair by homologous recombination. Consequently, loss of BRCA1 or BRCA2 function triggers replication arrest and accumulation of unrepaired DNA lesions. Enhancement of these phenotypes by DNA damaging chemo-therapeutic drugs such as cisplatin and PARP inhibitors underscores the hypersensitivity of BRCA1/2-deficient tumors to this type of treatments. In spite of the high rates of initial remission, most BRCA1/2-defective patients relapse and ultimately die of chemo-resistant disease. Therefore, there is a clear need to develop effective alternative therapies for this patient subset.

**Results**

We combined cellular models for BRCA1/2 inactivation with xenograft tumor growth in mice to demonstrate that exogenous acetaldehyde effectively and selectively reduced the survival of BRCA1/2-deficient cells and tumors including those that are PARP inhibitor-resistant. We further show that acetaldehyde perturbs replication speed and elicits DNA damage responses leading to apoptosis.

**Impact**

BRCA1 and BRCA2 prevent acetaldehyde-induced DNA damage and drugs that increase endogenous acetaldehyde levels (e.g., disulfiram) could be effective in treating BRCA1/2-compromised tumors.

polyclonal antibody raised against mouse BRCA2 (a gift from Dr. Hyunsook Lee, Seoul National University (Min *et al*, 2012)). Antibodies used for immunofluorescence detection were as follows: rabbit polyclonal antiserum raised against RAD51 (H92, Santa Cruz Biotechnology), rabbit polyclonal antiserum raised against RPA (SWE34, a gift from Dr. Steve West, Crick Institute, London), and mouse monoclonal antibody raised against phosphorylated histone H2AX Ser139 (clone JBW301, Merck Millipore).

**Data availability**

The authors declare that all relevant data supporting the findings of this study are available within the article and its supplementary information files, or from the corresponding author upon reasonable request.

**Expanded View** for this article is available online.

## Acknowledgements

We are grateful to K.J. Patel (LMB, Cambridge, UK) for providing the *Aldh2*$^{+/+}$ and *Aldh2*$^{-/-}$ MEFs and for insightful discussions. We thank Mick Woodcock and Graham Brown (CRUK/MRC Oxford Institute for Radiation Oncology, Oxford, UK) for their assistance with FACS and microscopy, respectively, as well as Steve West (CR-UK Clare Hall Laboratories) and Hyunsook Lee (Seoul National University, South Korea) for valuable reagents. Work in the A.B. laboratory is supported by Italian Association for Cancer Research (AIRC 16910). E.M.C.T. was supported by a Medical Research Council DPhil Studentship. G.Z. was supported by an Erasmus+ Mobility Scholarship. Work in M.T. laboratory is supported by Cancer Research UK, Medical Research Council, University of Oxford and EMBO Young Investigator Program.

## Author contributions

MT and EMCT designed the study and the experiments. MT, EMCT, and XL wrote the manuscript. EMCT and XL performed the majority of the experiments with the help of CF, GZ, SB, JM, and IS, VMG, PB, and JJ carried out clonogenic assays on mouse tumor-derived cell lines. MP and ABi carried out *in vivo* drug experiments, with assistance from AR, ASB, OMR, and ABr carried out *ex vivo* drug experiments. JH and NT provided the human fibroblast cell lines. MR and BR designed CRISPR constructs. All authors read and commented on the manuscript.

## Conflict of interest

The authors declare that they have no conflict of interest.

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
