## [Review Process File · EMBO Molecular Medicine]

BRCA1 and BRCA2 tumor suppressors protect against endogenous acetaldehyde toxicity

Eliana MC Tacconi, Xianning Lai, Cecilia Folio, Manuela Porru, Gijs Zonderland, Sophie Badie, Johanna Michl, Irene Sechi, Mélanie Rogier, Verónica Matía García, Ankita Sati Batra, Oscar M Rueda, Peter Bouwman, Jos Jonkers, Anderson Ryan, Bernardo Reina-San-Martin, Joannie Hui, Nelson Tang, Alejandra Bruna, Annamaria Biroccio, Madalena Tarsounas

Corresponding author: Madalena Tarsounas, University of Oxford

Review timeline:

Submission date:	08 December 2016
Editorial Decision:	25 January 2017
Additional Correspondence:	01 February 2017
Additional Correspondence:	02 February 2017
Resubmission:	12 May 2017
Editorial Decision:	13 June 2017
Revision Received	20 June 2017
Accepted:	22 June 2017

Transaction Report:

Editor: Roberto Buccione

1st Editorial Decision

25 January 2017

Thank you for the submission of your manuscript to EMBO Molecular Medicine. I apologise for the unusual delay in providing you with a decision, mostly due to difficulties in finding appropriate reviewers and then obtaining their evaluations in a timely manner over the holiday season.

We have now heard back from the three Reviewers whom we asked to evaluate your manuscript.

As you will see, the reviewers agree that the manuscript is of interest but they also raise fundamental issues. On one hand they note insufficient support for the findings and lack of robustness, on the other doubts are expressed on the overall significance and advance, and most importantly for our title, the actual translational implications based on the current experimentation.

After reviewer cross-commenting there was general agreement among all on the above concerns and also that they cannot be addressed within a reasonable timeframe.

Given therefore the above and the overall lack of enthusiasm from the reviewers, I have no choice but to return the manuscript to you at this stage with the message that we cannot offer publication of the manuscript in EMBO Molecular Medicine.

I am sorry to have to disappoint you at this stage. I hope that the reviewers' comments will be

helpful in your continued work in this area.

***** Reviewer's comments *****

Referee #1 (Remarks):

The Mspt of Tacconi et al, describes hypersensitivity of cells lacking BRCA2 to acetaldehyde due to toxic replication-associated DNA damage and thus suggest inhibitors of aldehyde metabolism (not acetaldehyde necessarily, see below) as selective means for elimination of BRCA1/2-deficient cells and tumors. As the main contribution, the study demonstrates that an existing drug, disulfiram, with relatively safe features (high and long term doses can generate Parkinson's disease like symptoms), is efficacious in treating BRCA2-deficient tumors. These observation is of a great significance from a clinical point of view.

The experimental part appears to support the conclusions, including the likelihood that the disulfiram effect can be attributed to inhibitions of ALDH rather than through other mechanisms. However, it is surprising that the majority of the data presented are from two independent experiments. Therefore, the SD values reflects technical variation rather than true biological replication.

Furthermore, I could not find how many mice were included in the study, if the data were statistically significant and if the data were scored blindly. Therefore, in my opinion, the mspt does not represent vigorous enough study.

the following points should also be addressed, at least in the discussion.

1. The effect of disulfiram is attributed by the authors to increase in endogenous acetaldehyde. However, no data are provided to substantiate this claim; the level of acetaldehyde or the levels of other aldehydes were not measured. A number of other endogenous aldehydes are metabolized by the disulfiram-sensitive ALDHs and their accumulation may have mediated the effect, so the claim needs to be rewarded.

2. The effect of acetaldehyde on DNA damage in vivo is well substantiated (in contrast to the second sentence in the discussion) and was the basis of classifying acetaldehyde as class I carcinogen by WHO. Furthermore, the role of acetaldehyde in early embryonic development and in what is called fetal alcohol syndrome is also well known and should not be attributed solely to the study of Langevin et al. 2011. Proper attribution to new comers to the field is important for the readership of the journal.

Referee #2 (Comments on Novelty/Model System):

BRCA1-deficient tumor cells only show modest effect of acetaldehyde in-vivo.

Referee #2 (Remarks):

The report suggests that BRCA1 and BRCA2 deficient cells are sensitive to acetaldehyde, extending the findings observed for Fanconi Anemia cells. The implications are that precipitating aldehyde toxicity may be a therapeutic strategy. The problem with that statement is that it is not clear whether therapeutic levels of aldehydes can be achieved in-vivo. The dose of acetaldehyde used in Fig.7 achieves some effect in BRCA2-deficient cells, but limited effect in BRCA1-deficient cells. The challenge is that although natural aldehyde production may be a source of genomic instability from low levels of DNA damage, converting this vulnerability into a therapeutic strategy may not be possible. As such, there is uncertainty about the full significance of the findings.

Having said this, there are interesting data presented, which are of interest. Although the toxicity data clearly show differences between BRCA1 or BRCA2-deficient cells and their wild-type controls, the survival data would be more meaningful with survival plotted on a logarithmic scale, preferably with a wider range of cell killing. For Figure 2B in particular, where the BRCA2-deficient cell data points are all along the baseline, the data should be replotted. The reasons for an upturn in the survival of cells by increasing the dose of disulfiram are not clear.

The replication track lengths are reduced in HR-defective cells and interestingly, FANCD2-deficient cells add to the defects observed with BRCA2-deficient cells. Survival however is not additive, but epistatic, which are interesting data but not fully explained or interpreted. The interpretation of Fig.

4 is most likely due to the profound chromosomal aberrations found in the BRCA2-deficient cells. Are signaling and G2/M differences all secondary to the chromosomal lesion?

Fig. 5 is interesting in that Aldh2^{-/-} cells are interactive with BRCA2-deficiency for impaired growth. The differences found for 53BP1 inactivation on PARP inhibitor sensitivity of BRCA1 deficient cells is not observed for acetaldehyde (Fig. 6C). This is a very interesting finding, but there is also no explanation of the finding? The RPA foci might have been interesting to observe in this context in addition to RAD51 (Fig 6D). If RPA foci are increased and RAD51 foci are decreased, resection may have been overcome, but not the ability to make RAD51. Why there should be differences between PARPi and acetaldehyde then becomes a very interesting question about the mechanism of how these different lesions engage the HR machinery.

Overall, these are interesting findings to the DNA repair specialists, but the full significance of the findings is not yet clear.

Referee #3 (Comments on Novelty/Model System):

From a basic science point of view the model lines are absolutely fine. From a clinical and translational perspective, patient derived human tumor material would be required.

Referee #3 (Remarks):

Tacconi and colleagues address the question, do BRCA1 and BRCA2 proteins protect against endogenous aldehyde toxicity. They report aldehyde sensitivity for cells knocked out and knocked down for BRCA2 and BRCA1, they show that inhibition of ALDH activity to impair aldehyde metabolism also sensitizes BRCA1 and BRCA2 deficient cells for killing. It is demonstrated that the ensuing cell death is linked to impaired DNA replication and results in the generation of ssDNA and DNA breaks. To confirm the contribution of HR to acetaldehyde toxicity they confirm that ALDH defective MEFs knocked down for different HR genes are impaired in cell growth. Lastly the authors report that acetaldehyde kills BRCA1 and BRCA2 deficient tumor cells.

The data presented in this manuscript are pretty solid and, overall, support the conclusions made.

There is a potential issue of novelty for discussion between editors and authors. At the beginning of the results section the authors indicate that FANCD2 cells are sensitive to acetaldehyde and this is linked to the latter's ability to inflict DNA crosslink damage. Since BRCA1 and BRCA2 are both in the FA pathway and are designated FA genes (FANCS and FANCD1 respectively) and human tumors lacking these genes are treated in the clinic with crosslinking agents, it is reasonably predicted that defects in these genes will also confer aldehyde sensitivity. The current manuscript does a good job of confirming this, but does not reveal any new mechanistic insight on how aldehyde damage is processed.

The work is good but whether this study is suited to EMBO Molecular Medicine is something for consideration.

There are a few minor comments-

1. The DLD1 BRCA2 ^{-/-} cells from Horizon Discovery should be cited in the main text to indicate clearly where they are from. Currently the citation occurs only in the Materials and Methods.
2. It would be helpful for the different type of chromosomal abnormalities caused by acetaldehyde treatment in BRCA2 cells to be reported in addition to an overall value. HR defective cells often exhibit increased numbers of radial chromosomes. If aldehyde treatment does not do this, it might be telling us something interesting.
3. The authors report elevated TP53 as part of their analysis of cell cycle signalling. P53 is mutated in DLD1 cells. The authors might comment on this in light of the fact that they reference increased P53 as a sign of normal ATM signaling and report that the checkpoint is normal in DLD1-BRCA2 deficient cells.

4. A small point- There is no need to color code the survival graphs and also label each graph with the color code. This can be done in the legend.

Additional Correspondence

01 February 2017

Thank you very much for sending us the reviews of our paper.

We agree with some of the points raised by the Reviewers and have already set up experiments to address them. In addition, several concerns can be clarified in the text in a revised version of our manuscript. The key point raised by all Reviewers seems to be unsubstantiated clinical relevance of our work, hence the suggestion to validate our results using patient-derived tumour xenograft (PDX) models. To this end, we are collaborating with Dr. Alejandra Bruna in Carlos Caldas group at the CRUK Cambridge Institute, who is currently testing acetaldehyde and disulfiram using PDX ex vivo models. Establishment of PDX-derived tumour cells (PDTCs) was described in her recent paper (attached). These are short term cultures established immediately after tumour extraction, that recapitulate tumour heterogeneity and drug responses. Thus, PDTCs represent a time- and cost-effective surrogate for in vivo PDX experiments. I wonder whether with these experimental additions (and others on which I will not elaborate here) you would reconsider our paper for publication.

Many thanks in advance and I look forward to your reply.

Additional Correspondence

02 February 2017

What you propose appears sound and with great potential to substantially improve the manuscript according to the reviewers comments. We would thus be happy to reconsider a manuscript revised along the lines you suggest. This would be a new submission however and although I can commit to at least attempting to secure the same reviewers, this might not be possible.

I look forward to reading your revised manuscript in due time.

Resubmission

12 May 2017

Referee #1:

The Mspt of Tacconi et al, describes hypersensitivity of cells lacking BRCA2 to acetaldehyde due to toxic replication-associated DNA damage and thus suggest inhibitors of aldehyde metabolism (not acetaldehyde necessarily, see below) as selective means for elimination of BRCA1/2-deficient cells and tumors. As the main contribution, the study demonstrates that an existing drug, disulfiram, with relatively safe features (high and long term doses can generate Parkinson's disease like symptoms), is efficacious in treating BRCA2-deficient tumors. These observation is of a great significance from a clinical point of view.

The experimental part appears to support the conclusions, including the likelihood that the disulfiram effect can be attributed to inhibitions of ALDH rather than through other mechanisms.

However, it is surprising that the majority of the data presented are from two independent experiments. Therefore, the SD values reflects technical variation rather than true biological replication.

Response: The viability results are presented in our paper as two independent repeats, with three technical replicates each. These results have been reproduced using clonogenic analyses of cell survival (shown in Supplementary Data) in at least in two more independent experiments with three technical replicates. Therefore, the validity of our data is supported by robust experimental evidence.

Furthermore, I could not find how many mice were included in the study, if the data were statistically significant and if the data were scored blindly. Therefore, in my opinion, the mspt does not represent vigorous enough study.

Response: We have stated that 'Each experimental group included five mice' both in Materials and Methods (p. 21; middle) and legend to Figure 7 (p. 32). The statistical significance of our results, discussed in the Results section (p. 11) and shown in Figure 7 legend (p. 32), emphasizes the robustness of our data and the strong support they provide to the conclusions of our paper.

The following points should also be addressed, at least in the discussion.

1. The effect of disulfiram is attributed by the authors to increase in endogenous acetaldehyde. However, no data are provided to substantiate this claim; the level of acetaldehyde or the levels of other aldehydes were not measured. A number of other endogenous aldehydes are metabolized by the disulfiram-sensitive ALDHs and their accumulation may have mediated the effect, so the claim needs to be rewarded.

Response: We agree with the Referee that ALDHs are known to metabolize other endogenous aldehydes and we emphasize this in the second sentence of Discussion (p. 13). However, our paper is focusing on acetaldehyde toxicity to BRCA1/2-deficient cells and tumors and on ALDH2, as the best characterized enzyme required for processing acetaldehyde to acetate. The rationale for our approach is stated in Results (p. 6). Characterization of the effects of other aldehydes in our model system is beyond the scope of the current paper and represents the focus of our future work.

2. The effect of acetaldehyde on DNA damage *in vivo* is well substantiated (in contrast to the second sentence in the discussion) and was the basis of classifying acetaldehyde as class I carcinogen by WHO. Furthermore, the role of acetaldehyde in early embryonic development and in what is called fetal alcohol syndrome is also well known and should not be attributed solely to the study of Langevin *et al.* 2011. Proper attribution to new comers to the field is important for the readership of the journal.

Response: The Referee is correct in stating that the DNA damaging effects of acetaldehyde *in vivo* have been documented in several publications, which now have been included in the revised text (Seitz and Stickel, *Nat. Rev. Cancer* 2007; Brooks and Theruvathu, *Alcohol* 2005; Brooks and Zakhari, *Environ. Mol. Mutagen.* 2014). However, the key role of ALDH2 enzyme in the repair of this damage only emerged recently from the work of KJ Patel laboratory (LMB, Cambridge). We have amended the fourth sentence of Discussion (p. 13) to incorporate this correction and the relevant references.

Referee #2:

(Comments on Novelty/Model System): BRCA1-deficient tumor cells only show modest effect of acetaldehyde *in-vivo*.

Response: Our data shown in Fig 7B and C indicate that *Brcal*^{-/-} allograft tumors are sensitive to acetaldehyde treatment. In *Brcal*^{-/-} olaparib-sensitive tumors (Fig 7B), TWI values in response to PARP inhibitor and acetaldehyde are 40% and 44%, respectively. In *Brcal*^{-/-} olaparib-resistant tumors (Fig 7C), TWI in response to acetaldehyde is 54%. In both cases, the response to acetaldehyde treatment was superior to that to PARP inhibitor, which is currently used in the clinic as the compound with highest specificity against BRCA1/2-deleted tumors. Based on these observations and the *P* values for statistical significance included in the revised version of our manuscript (Results, p. 11 and Figure legends, p. 32), we concluded that the effects observed *in vivo* are significant and support the conclusions of our paper.

(Remarks): The report suggests that BRCA1 and BRCA2 deficient cells are sensitive to acetaldehyde, extending the findings observed for Fanconi Anemia cells. The implications are that precipitating aldehyde toxicity may be a therapeutic strategy. The problem with that statement is that it is not clear whether therapeutic levels of aldehydes can be achieved *in-vivo*. The dose of acetaldehyde used in Fig.7 achieves some effect in BRCA2-deficient cells, but limited effect in BRCA1-deficient cells. The challenge is that although natural aldehyde production may be a source of genomic instability from low levels of DNA damage, converting this vulnerability into a therapeutic strategy may not be possible. As such, there is uncertainty about the full significance of the findings.

Having said this, there are interesting data presented, which are of interest. Although the toxicity data clearly show differences between BRCA1 or BRCA2-deficient cells and their wild-type controls, the survival data would be more meaningful with survival plotted on a logarithmic scale, preferably with a wider range of cell killing. For Figure 2B in particular, where the BRCA2-

deficient cell data points are all along the baseline, the data should be replotted. The reasons for an upturn in the survival of cells by increasing the dose of disulfiram are not clear.

Response: We would like to thank the Referee for the suggestion to plot our data in Fig. 2B on logarithmic scale. The new graph prepared accordingly is included in the revised version of our manuscript. We also agree with the Referee that the upturn in cell survival by increasing disulfiram concentration is an important and intriguing observation. Although we could not address it experimentally, we speculate on p. 6 (bottom) of our revised manuscript, that at high concentrations disulfiram may form aggregates which cannot penetrate the cell membrane. This could explain the lower efficiency of the drug at high concentrations.

The replication track lengths are reduced in HR-defective cells and interestingly, FANCD2-deficient cells add to the defects observed with BRCA2-deficient cells. Survival however is not additive, but epistatic, which are interesting data but not fully explained or interpreted.

Response: Indeed, as the Referee points out nascent replication tracks are shorter in HR-defective cells (Fig 3D). However, the effects observed in FANCD2-deficient cells are comparable, not additive, to the effect observed in HR-deficient cells. Consistent with this, the impact on survival is also similar in HR- and FA-defective cells. We have explained these results in p. 7 (bottom) of Results and proposed an interpretation on p. 13 (bottom) of Discussion.

The interpretation of Fig. 4 is most likely due to the profound chromosomal aberrations found in the BRCA2-deficient cells. Are signaling and G2/M differences all secondary to the chromosomal lesion?

Response: Our assay lacks the resolution necessary to determine the temporal succession of these events. Using Western blot analyses of samples collected at various time points, we could detect DNA lesions occurring concomitantly with DNA damage signaling activation.

Fig. 5 is interesting in that Aldh2-/- cells are interactive with BRCA2-deficiency for impaired growth. The differences found for 53BP1 inactivation on PARP inhibitor sensitivity of BRCA1 deficient cells is not observed for acetaldehyde (Fig. 6C). This is a very interesting finding, but there is also no explanation of the finding? The RPA foci might have been interesting to observe in this context in addition to RAD51 (Fig 6D). If RPA foci are increased and RAD51 foci are decreased, resection may have been overcome, but not the ability to make RAD51. Why there should be differences between PARPi and acetaldehyde then becomes a very interesting question about the mechanism of how these different lesions engage the HR machinery.

Overall, these are interesting findings to the DNA repair specialists, but the full significance of the findings is not yet clear.

Response: We would like to thank the Referee for suggesting to monitor the RPA focus formation in response to PARPi and acetaldehyde. We attempted this experiment, however failed to identify an antibody that recognized mouse RPA in immunofluorescence assays. We agree that determining the interplay between RAD51 and RPA foci would provide important mechanistic insights into the repair of lesions inflicted by these treatments. We therefore plan to make this line of investigation a focus of our future studies.

Referee #3:

(Comments on Novelty/Model System): From a basic science point of view the model lines are absolutely fine. From a clinical and translational perspective, patient derived human tumor material would be required.

Response: We would like to thank the Referee for this comment. In order to increase the clinical and translational relevance of our work, we have added two new lines of experimentation to our revised manuscript:

1. We have addressed the toxicity of disulfiram to patient-derived tumor xenografts cells (PDTCs). These are short-term cultures established from human breast and metastatic tumors which recapitulate tumor heterogeneity and responses to therapy in culture. These models for BRCA1-deficient tumors have been recently reported (Bruna et al., *Cell* 2016). Our results (included in new Fig. 7F) indicate that disulfiram is toxic to BRCA1-deficient, but not BRCA1-proficient PDTCs, thus confirming our model that HR-deficient cells and tumors can be effectively eliminated using this drug (p. 11-12 of Results and p. 15 of Discussion).
2. We have included in our analysis human fibroblasts obtained from colleagues in Hong Kong, which carry the naturally occurring rs671 ALDH2 mutation. A single G to A nucleotide change causes the substitution of glutamate to lysine at position 487 (E487K). In

homozygous carriers, this substitution is associated with the ethanol-induced flushing syndrome (Yoshida et al., *PNAS* 1984). Using cell lines derived from *ALDH2* mutant and wild type individuals, we show that HR inactivation through depletion of *RAD51*, *BRCA1* or *BRCA2* specifically decreases proliferation of *ALDH2* mutant cells. These results corroborate our data obtained in mouse embryonic fibroblasts carrying *Aldh2* gene deletion. Given that approximately 560 million East Asians (8% of the world population) carry this mutant allele (Chen et al., *Physiol. Rev* 2014), our results gain a substantial translational relevance because they predict a low occurrence of *BRCA2*-mutated breast and ovarian cancers in this population. These data are now presented on p. 10 and Fig. S9 of our revised manuscript.

(Remarks): Tacconi and colleagues address the question, do BRCA1 and BRCA2 proteins protect against endogenous aldehyde toxicity. They report aldehyde sensitivity for cells knocked out and knocked down for BRCA2 and BRCA1, they show that inhibition of ALDH activity to impair aldehyde metabolism also sensitizes BRCA1 and BRCA2 deficient cells for killing. It is demonstrated that the ensuing cell death is linked to impaired DNA replication and results in the generation of ssDNA and DNA breaks. To confirm the contribution of HR to acetaldehyde toxicity they confirm that ALDH defective MEFs knocked down for different HR genes are impaired in cell growth. Lastly the authors report that acetaldehyde kills BRCA1 and BRCA2 deficient tumor cells. The data presented in this manuscript are pretty solid and, overall, support the conclusions made. There is a potential issue of novelty for discussion between editors and authors. At the beginning of the results section the authors indicate that FANCD2 cells are sensitive to acetaldehyde and this is linked to the latter's ability to inflict DNA crosslink damage. Since BRCA1 and BRCA2 are both in the FA pathway and are designated FA genes (FANCS and FANCD1 respectively) and human tumors lacking these genes are treated in the clinic with crosslinking agents, it is reasonably predicted that defects in these genes will also confer aldehyde sensitivity. The current manuscript does a good job of confirming this, but does not reveal any new mechanistic insight on how aldehyde damage is processed.

The work is good but whether this study is suited to EMBO Molecular Medicine is something for consideration.

Response: Whether *BRCA1* and *BRCA2* belong to the Fanconi anaemia pathway is an ongoing controversy in the field, aspects of which were discussed in our recent review by Michl et al. (*EMBO J* 35, 2016). *BRCA1* and *BRCA2* are designated FA genes based on a very limited number of patients (only 2 for *BRCA1*). *BRCA1* and *BRCA2* mutations predispose to breast and ovarian tumors, whilst FA patients never get these types of cancers. There is extensive cross talk between the FA and HR repair genes and proteins, but they are by no means equivalent. Indeed, as the Referee points out, *BRCA1/2*-defective breast and ovarian tumors are treated in the clinic with crosslinking agents. However, one cannot simply predict that they will respond similarly to acetaldehyde exposure, as there are examples of crosslinking agents which are not effective against *BRCA1/2* deficiencies. Most relevant to the novelty of our results, is that acetaldehyde is a product of endogenous cellular metabolism. As such, it represents an intrinsic source of DNA damage, particularly toxic in the context of *BRCA1/2*-deficiency. Consistent with this are our results that human and mouse cells lacking *BRCA1/2* and *ALDH2* show impaired proliferation (Fig 5 and Fig S9), suggestive of a synthetic lethal interaction between these genes.

There are a few minor comments-

1. The DLD1 BRCA2 -/- cells from Horizon Discovery should be cited in the main text to indicate clearly where they are from. Currently the citation occurs only in the Materials and Methods.

Response: We have included the source of DLD1 *BRCA2*^{-/-} cells both in Results (p. 5) and Materials and Methods (p. 16).

2. It would be helpful for the different type of chromosomal abnormalities caused by acetaldehyde treatment in BRCA2 cells to be reported in addition to an overall value. HR defective cells often exhibit increased numbers of radial chromosomes. If aldehyde treatment does not do this, it might be telling us something interesting.

Response: The radial chromosomes were detected at very low frequencies in our analyses. The chromosome abnormalities consisted predominantly of chromatid and chromosome breaks.

3. The authors report elevated TP53 as part of their analysis of cell cycle signalling. P53 is mutated in DLD1 cells. The authors might comment on this in light of the fact that they reference increased P53 as a sign of normal ATM signaling and report that the checkpoint is normal in DLD1-BRCA2 deficient cells.

Response: The Reviewer is correct in that DLD1 cells harbor a p53 mutation. In our unpublished data, we observed that p53 expression and Ser15-p53 levels are remarkably elevated in these cells, both in response to BRCA2 abrogation and to IR treatment. However, p21 expression is not increased, which reflects p53 pathway deregulation. To eliminate confusion, we have excluded the p53 induction data from our revised manuscript.

4. *A small point- There is no need to color code the survival graphs and also label each graph with the color code. This can be done in the legend.*

Response: We would like to thank the Referee for this helpful suggestion. We will remove color in the final version of our manuscript.

2nd Editorial Decision

13 June 2017

Thank you for the submission of your manuscript to EMBO Molecular Medicine and apologies for the delay in providing you with a decision due to the difficulties in obtaining the reviewer evaluations in a timely manner.

We have now received comments from the two out of the three Reviewers whom we asked to evaluate your manuscript. Unfortunately, we have been unable to contact reviewer 3.

As a further delay cannot be justified and would not be productive, I decided to proceed based on the two available evaluations. As you will see, the two reviewers confirm the interest of your study but especially reviewer 1, do not appear very enthusiastic. Reviewer 2 suggests that perhaps the study is better suited to a specialist readership at this stage of development.

Given the fact that reviewer 3 was not available to evaluate your rebuttal, I followed up by seeking further independent advice from a very expert external advisor. The advisor felt that the work was well-performed and with no significant flaws, and that the data support the main conclusions, i.e. that BRCA1/2 protect against acetaldehyde toxicity. We also agreed that your work, although somewhat similar to a very recently published Cell paper (PMID:28575672), is complementary, provides needed support and is important for the field.

In conclusion, and following further internal discussion, I am pleased to inform you that we will be able to accept your manuscript pending the successful completion of the following final amendments:

- 1) Please carefully respond to the concerns raised by reviewers 1 and 2 by providing a point-by-point rebuttal and appropriate textual changes in the manuscript.
- 2) Please provide individual figure files as per our Author Guidelines.
- 3) Please provide a running title, a conflict of interest statement and the "Author Contributions" and "The Paper Explained" sections as per our Author Guidelines.
- 4) Please list reference in the 20 authors et al. format (they are currently listed as 10 authors et al.
- 5) Please rename the supplementary figures and tables as per our Author Guidelines (<http://embomolmed.embopress.org/authorguide#expandedview>) and consequently amend relative manuscript callouts. In brief, this information should be provided as a single Appendix PDF and the nomenclature to name and refer to Appendix items in the main text is: Appendix Figure S1, Appendix Table S1, Appendix Supplementary Methods, etc.). Also, the Appendix should begin with a short table of contents. When published, the Appendix will be provided at the end of the manuscript as a PDF download.
- 6) Please provide scale bars for Figs 3A and S3. We also note that Fig. 6E is excessively contrasted, please provide better images and also refer to item 10 below.
- 7) As per our Author Guidelines, the description of all reported data that includes statistical testing must state the name of the statistical test used to generate error bars and P values, the number (n) of independent experiments underlying each data point (not replicate measures of one sample), and the actual P value for each test (not merely 'significant' or 'P < 0.05'). If you wish to collect the P values

together separately to avoid figure clutter, you may prepare an additional supplementary table, which of course should have appropriate callouts in the main text

8) Please note that EMBO Molecular Medicine now requires a complete author checklist (<http://embomolmed.embopress.org/authorguide#editorial3>) to be submitted with all revised manuscripts. Provision of the author checklist is mandatory at revision stage; The checklist is designed to enhance and standardize reporting of key information in research papers and to support reanalysis and repetition of experiments by the community. The list covers key information for figure panels and captions and focuses on statistics, the reporting of reagents, animal models and human subject-derived data, as well as guidance to optimise data accessibility. The Author checklist will be published alongside the paper, in case of acceptance, within the transparent review process file.

9) In connection to the above, the manuscript must include a statement in the Materials and Methods identifying the institutional and/or licensing committee approving the experiments, including any relevant details (like how many animals were used, of which gender, at what age, which strains, if genetically modified, on which background, housing details, etc). We encourage authors to follow the ARRIVE guidelines for reporting studies involving animals. Please see the EQUATOR website for details: <http://www.equator-network.org/reporting-guidelines/improving-bioscience-research-reporting-the-arrive-guidelines-for-reporting-animal-research/>. Please make sure that ALL the above details are reported.

10) We encourage the publication of source data, with the aim of making primary data more accessible and transparent to the reader. Would you be willing to provide a PDF file per figure that contains the original, uncropped and unprocessed scans of all or at least the key gels used in the manuscript and/or source data sets for relevant graphs? The files should be labeled with the appropriate figure/panel number, and in the case of gels, should have molecular weight markers; further annotation may be useful but is not essential. The files will be published online with the article as supplementary "Source Data" files. If you have any questions regarding this just contact me.

11) Every published paper includes a 'Synopsis' to further enhance discoverability. Synopses are displayed on the journal webpage and are freely accessible to all readers. They include a short description as well as 2-5 one-sentence bullet points that summarise the key NEW findings of the paper. The bullet points should be designed to be complementary to the abstract - i.e. not repeat the same text. We encourage inclusion of key acronyms and quantitative information. Please use the passive voice. Please attach this information in a separate file or send them by email, we will incorporate it accordingly. We also encourage the provision of striking image or visual abstract to illustrate your article. If you do, please provide a jpeg file 550 px-wide x 400-px high.

I look forward to reading your next, final version of your manuscript as soon as possible.

***** Reviewer's comments *****

Referee #1 (Remarks):

I read the revised manuscript, and the response of the authors to the comments. As indicated in my first review, I view the study interesting, but keep my rating for the technical quality as low. The manuscript reports data from only one (representative according to the authors) of only two experiments, showing technical triplicates - which is far from being rigorous enough; biological replication of independent experiments should be provided. The authors could have at least provided the data from the other experiment, to allow the reader to confirm that indeed they are representative data. Furthermore, the fact that there is statistical significance in an in vivo study with each group having five mice is insufficiently rigorous. Again, I find the study interesting and important, but am reluctant to suggest its acceptance for publication for the above reason

Referee #2 (Remarks):

The revised paper and rebuttal to the previous reviews suggest that the authors have attempted to answer the issues as far as is easily possible. The question is whether the response is satisfactory - in summary they have answered what they could in the time available, before the results become outdated by other reports.

The concern about whether aldehyde toxicity may be a therapeutic strategy has not really been addressed. There remains uncertainty about the full significance of these findings. The additivity of replication fork track lengths in FANCD2-deficient and BRCA2-deficient cells does not correlate with survival, which is not additive. Others have reported (only in meetings, not yet published) synthetic lethality between BRCA2 and FANCD2. This section remains confusing and unclear.

Overall, the responses to the previous reviews are reasonable, but not comprehensive. For a journal with an impact factor of ~9.5, this paper is unlikely to improve the rating. However, to the DNA repair field, this report is of interest.

1st Revision

20 June 2017

Referee #1 (Remarks):

I read the revised manuscript, and the response of the authors to the comments. As indicated in my first review, I view the study interesting, but keep my rating for the technical quality as low. The manuscript reports data from only one (representative according to the authors) of only two experiments, showing technical triplicates - which is far from being rigorous enough; biological replication of independent experiments should be provided. The authors could have at least provided the data from the other experiment, to allow the reader to confirm that indeed they are representative data. Furthermore, the fact that there is statistical significance in an in vivo study with each group having five mice is insufficiently rigorous. Again, I find the study interesting and important, but am reluctant to suggest its acceptance for publication for the above reason.

Response: We would like to thank the reviewer for acknowledging the importance of our study. We have now repeated our viability assays at least three times. We show representative data from one experiment as viability assays carry an inherent level of variability and therefore results from one experiment (with triplicate technical replicates) are shown routinely in the literature (Boersma et al., 2015; Evers et al., 2008, 2010; Zimmer et al., 2016). The use of five mice per experimental group is acceptable practice, in line with national and international guidelines for Laboratory Animal Care and as in previous publications (Zimmer et al., 2016).

Referee #2 (Remarks):

The revised paper and rebuttal to the previous reviews suggest that the authors have attempted to answer the issues as far as is easily possible. The question is whether the response is satisfactory - in summary they have answered what they could in the time available, before the results become outdated by other reports. The concern about whether aldehyde toxicity may be a therapeutic strategy has not really been addressed. There remains uncertainty about the full significance of these findings. The additivity of replication fork track lengths in FANCD2-deficient and BRCA2-deficient cells does not correlate with survival, which is not additive. Others have reported (only in meetings, not yet published) synthetic lethality between BRCA2 and FANCD2. This section remains confusing and unclear.

Overall, the responses to the previous reviews are reasonable, but not comprehensive. For a journal with an impact factor of ~9.5, this paper is unlikely to improve the rating. However, to the DNA repair field, this report is of interest.

Response: We agree with the Reviewer that the genotoxic effects of aldehydes may prevent them from becoming a therapeutic strategy. However, modulating the levels of endogenous aldehyde could be exploited clinically. We stated this on p. 15 (bottom) of the revised manuscript. Contrary to the Reviewer's statement, the synthetic lethality between BRCA2 and FANCD2 genes under physiological conditions has been published last year by our group (Michl et al., *NSMB* 2016) and Alan D'Andrea's group (Kais et al., *Cell Reports* 2016). On p. 13 (bottom), we explain that our

results support an epistatic interaction between the two genes in response to acetaldehyde treatment, distinct from the unchallenged conditions.

2) Please provide individual figure files as per our Author Guidelines.

Response: Individual figure files have been provided.

3) Please provide a running title, a conflict of interest statement and the "Author Contributions" and "The Paper Explained" sections as per our Author Guidelines.

Response: Running title, conflict of interest statement, author contributions and "The Paper Explained" have been provided.

4) Please list reference in the 20 authors et al. format (they are currently listed as 10 authors et al.

Response: The references have been reformatted to reflect 20 authors et al.

5) Please rename the supplementary figures and tables as per our Author Guidelines

(<http://embomolmed.embopress.org/authorguide#expandedview>) and consequently amend relative manuscript callouts. In brief, this information should be provided as a single Appendix PDF and the nomenclature to name and refer to Appendix items in the main text is: Appendix Figure S1, Appendix Table S1, Appendix Supplementary Methods, etc.). Also, the Appendix should begin with a short table of contents. When published, the Appendix will be provided at the end of the manuscript as a PDF download.

Response: All supplementary figures have been renamed to adhere to the author guidelines.

6) Please provide scale bars for Figs 3A and S3. We also note that Fig. 6E is excessively contrasted, please provide better images and also refer to item 10 below.

Response: The scale bars for Fig 3A and S3 are now included. The contrast of the western blot in Fig 4E (not 6E) has now been adjusted.

7) As per our Author Guidelines, the description of all reported data that includes statistical testing must state the name of the statistical test used to generate error bars and P values, the number (n) of independent experiments underlying each data point (not replicate measures of one sample), and the actual P value for each test (not merely 'significant' or 'P < 0.05'). If you wish to collect the P values together separately to avoid figure clutter, you may prepare an additional supplementary table, which of course should have appropriate callouts in the main text

Response: Actual P values have been included in the figures.

8) Please note that EMBO Molecular Medicine now requires a complete author checklist (<http://embomolmed.embopress.org/authorguide#editorial3>) to be submitted with all revised manuscripts. Provision of the author checklist is mandatory at revision stage; The checklist is designed to enhance and standardize reporting of key information in research papers and to support reanalysis and repetition of experiments by the community. The list covers key information for figure panels and captions and focuses on statistics, the reporting of reagents, animal models and human subject-derived data, as well as guidance to optimise data accessibility. The Author checklist will be published alongside the paper, in case of acceptance, within the transparent review process file.

Response: A complete author checklist is included.

9) In connection to the above, the manuscript must include a statement in the Materials and Methods identifying the institutional and/or licensing committee approving the experiments, including any relevant details (like how many animals were used, of which gender, at what age, which strains, if genetically modified, on which background, housing details, etc). We encourage authors to follow the ARRIVE guidelines for reporting studies involving animals. Please see the EQUATOR website for details: <http://www.equator-network.org/reporting-guidelines/improving-bioscience-research-reporting-the-arrive-guidelines-for-reporting-animal-research/>. Please make sure that ALL the above details are reported.

Response: This information is included in the Materials and Methods section.

10) We encourage the publication of source data, with the aim of making primary data more accessible and transparent to the reader. Would you be willing to provide a PDF file per figure that

contains the original, uncropped and unprocessed scans of all or at least the key gels used in the manuscript and/or source data sets for relevant graphs? The files should be labeled with the appropriate figure/panel number, and in the case of gels, should have molecular weight markers; further annotation may be useful but is not essential. The files will be published online with the article as supplementary "Source Data" files. If you have any questions regarding this just contact me.

Response: Original, uncropped and unprocessed scans of key western blots in the main figures are provided as Source Data files.

11) Every published paper includes a 'Synopsis' to further enhance discoverability. Synopses are displayed on the journal webpage and are freely accessible to all readers. They include a short description as well as 2-5 one-sentence bullet points that summarise the key NEW findings of the paper. The bullet points should be designed to be complementary to the abstract - i.e. not repeat the same text. We encourage inclusion of key acronyms and quantitative information. Please use the passive voice. Please attach this information in a separate file or send them by email, we will incorporate it accordingly. We also encourage the provision of striking image or visual abstract to illustrate your article. If you do, please provide a jpeg file 550 px-wide x 400-px high.

Response: A synopsis for the manuscript is included.

YOU MUST COMPLETE ALL CELLS WITH A PINK BACKGROUND ↓
PLEASE NOTE THAT THIS CHECKLIST WILL BE PUBLISHED ALONGSIDE YOUR PAPER

Corresponding Author Name: Madalena Tarsounas
Journal Submitted to: EMBO Molecular Medicine
Manuscript Number: EMM-2016-07446-V2